# DisCo-DSO: Coupling Discrete and Continuous Optimization for Efficient Generative Design in Hybrid Spaces

## Abstract

In this paper, we consider the challenge of optimizing within hybrid discrete-continuous spaces, a problem that arises in various important applications, such as symbolic regression and decision tree learning. We propose DisCo-DSO (Discrete-Continuous Deep Symbolic Optimization), a novel approach that uses a generative model to learn a joint distribution over discrete and continuous design variables to sample new hybrid designs. In contrast to standard decoupled approaches, in which the discrete and continuous variables are optimized separately, our joint optimization approach uses fewer objective function evaluations, is robust against non-differentiable objectives, and learns from prior samples to guide the search, which leads to significant improvement in performance and efficiency. Our experiments on a diverse set of optimization tasks demonstrate that the advantages of DisCo-DSO become increasingly evident as problem complexity grows. In particular, we illustrate DisCo-DSO's superiority over the state-of-the-art methods for interpretable reinforcement learning with decision trees.

## 1 Introduction

Deep learning methods have demonstrated success on important combinatorial optimization problems (Bello et al., 2016), including symbolic regression (SR) for discovering underlying mathematical equations from data (Petersen et al., 2021a; Biggio et al., 2021; Kamienny et al., 2022), and generating interpretable policies for continuous control (Landajuela et al., 2021c). Existing approaches train a generative model that constructs a solution to the optimization problem by sequentially choosing from a set of discrete tokens, using the objective function value as the terminal reward for learning. However, these approaches do not fully account for the hybrid discrete-continuous nature of such optimization problems: certain discrete tokens require the additional specification of an associated real-valued parameter, such as the value of a constant token in an equation or the threshold value at a decision tree node, but the learned generative model does not produce these values. Instead, they adopt the design choice of decoupled optimization, whereby only the construction of a discrete solution skeleton is optimized by deep learning while the associated continuous parameters are left to a separate black-box optimizer.

We hypothesize that a joint discrete-continuous optimization approach (Figure 1b) that generates a complete solution based on deep reinforcement learning (RL) (Sutton & Barto, 2018) has significant advantages compared to existing decoupled approaches that employ learning only for the discrete skeleton (Figure 1a). In terms of efficiency, a joint approach only requires one evaluation of the objective function for each solution candidate, whereas the decoupled approach based on common nonlinear black-box optimization methods such as BFGS (Fletcher, 2000), simulated annealing (Xiang et al., 1997), or evolutionary algorithms (Storn & Price, 1997) requires a significant number of function evaluations to complete each discrete skeleton. This incurs a high cost for applications such as interpretable control, where each evaluation involves running the candidate solution on many episodes of a high-dimensional and stochastic physical simulation (Landajuela et al., 2021c). Furthermore, joint exploration and learning on the full discrete-continuous solution space has the potential to escape from local optima and use information from prior samples to guide the subsequent search. In contrast, the decoupled approach must optimize continuous parameters from scratch for every discrete skeleton, and the lack of exploration in certain classes of optimizers means that the

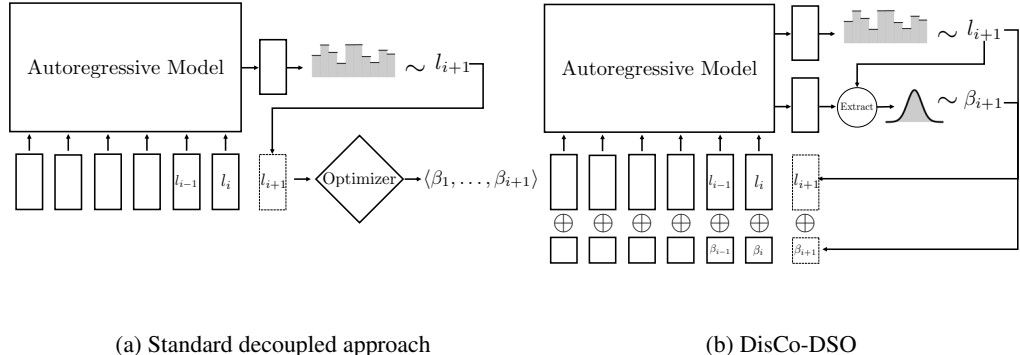

(a) Standard decoupled approach      (b) DisCo-DSO

Figure 1: Comparison of the standard decoupled approach and DisCo-DSO. In the decoupled approach, the discrete skeleton $\tau_d$ is sampled first and then the continuous parameters $\beta_1, \ldots, \beta_T$ are optimized independently. In contrast, DisCo-DSO models the joint distribution over the sequence of tokens $\langle (l_1, \beta_1), \ldots, (l_T, \beta_T) \rangle$. Here, the notation $\oplus$ stands for concatenation of vectors.

continuous parameters may be suboptimal for highly nonlinear or discontinuous objective functions. That would result in poor and misleading rewards for the generative policy even if it produced the correct discrete skeleton.

In this paper, we draw upon the success of deep reinforcement learning in parameterized action space Markov decision processes (Hausknecht & Stone, 2016) to extend existing deep learning methods for discrete optimization (Bello et al., 2016; Zoph & Le, 2017; Petersen et al., 2021a; Landajuela et al., 2021c) to the broader space of joint discrete-continuous optimization. Our general method for joint optimization, which we call DisCo-DSO (Discrete-Continuous Deep Symbolic Optimization), outperforms previous decoupled approaches that invoke widely-used standalone optimizers for the continuous parameters. Using a pedagogical parameterized bitstring task with known global optima, we show that our approach overcomes difficult nonlinearities and discontinuities whereas the decoupled approach requires either significantly more samples or fails to escape local optima. We further show that the advantage of DisCo-DSO over decoupled approaches increases with problem complexity, by evaluating on symbolic regression benchmarks with large number of constants. Finally, we use DisCo-DSO to generate interpretable decision tree (DT) policies for standard control benchmarks in reinforcement learning, outperforming DT-specific methods on all environments.

## 2 RELATED WORK

In the field of symbolic regression, different approaches have been proposed for addressing the optimization of both discrete skeletons and continuous parameters. Traditional genetic programming approaches and deep generative models handle these problems separately, with continuous constants optimized after discrete parameters (Topchy et al., 2001; Petersen et al., 2021a; Biggio et al., 2021). Recent work aim to jointly optimize discrete constants and continuous parameters by relaxing the discrete problem into a continuous one (Martius & Lampert, 2016; Sahoo et al., 2018), or tokenizing (i.e., discretizing) the continuous constants (Kamienny et al., 2022). The former approach faces challenges such as gradient blow-ups and the need to revert continuous values to discrete ones. The latter approach tokenizes continuous constants, treating them similarly to discrete tokens, but such quantization is problem-dependent, restricts the search space, and requires additional post-hoc optimization to refine the continuous parameters. In contrast, our work addresses these limitations by proposing a general method that treats both discrete tokens and continuous parameters on an equal footing and optimizes them jointly, without needing problem-specific conversions.

In the domain of symbolic reinforcement learning, where the goal is to find intelligible and concise control policies, works such as Landajuela et al. (2021c) and Sahoo et al. (2018) have used the continuous-tokenization and relaxation approaches, respectively, to optimize symbolic control

policies in continuous action spaces. For discrete action spaces, the natural representation of the policy is a DT (Ding et al., 2020; Silva et al., 2020; Custode & Iacca, 2023). In Custode & Iacca (2023), the authors use an evolutionary search to find the best DT policy and further optimized the real valued thresholds using a decoupled approach. Relaxation approaches find their counterparts within this domain in works such as (Sahoo et al., 2018; Silva et al., 2020; Ding et al., 2020), where a soft DT is used to represent the policy. The soft DT, which fixes the discrete structure of the policy and exposes the continuous parameters, is then optimized using gradient based methods.

The treatment of the continuous parameters as part of the action space has strong parallels in the space of hybrid discrete-continuous RL. In Hausknecht & Stone (2016), the authors present a successful application of deep reinforcement learning to a domain with continuous state and action spaces. In Xiong et al. (2018), the authors take an off-policy DQN-type approach that directly works on the hybrid action space without approximation of the continuous part or relaxation of the discrete part, but requires an extra loss function for the continuous actions. In Neunert et al. (2020), they propose a hybrid RL algorithm that uses continuous policies for discrete action selection and discrete policies for continuous action selection. Similar to these works, we treat the continuous parameters as part of the action space and optimize them jointly with the discrete tokens. However, our approach distinguishes itself from these prior works by customizing both the problem definition and the optimization loss to suit the discrete-continuous optimization problem. As a matter of fact, our paper represents, to the best of our knowledge, the first known instance of using deep reinforcement learning in parameterized discrete-continuous action spaces for the purpose of discrete optimization.

## 3 DISCRETE-CONTINUOUS DEEP SYMBOLIC OPTIMIZATION

### 3.1 NOTATION

We consider a discrete-continuous optimization problem defined over a search space $\mathcal{T}$ of sequences of tokens $\tau = \langle \tau_1, \ldots, \tau_T \rangle$, where each token $\tau_i, \forall i \in \{1, \ldots, T\}$, belongs to a library $\mathcal{L}$. The library $\mathcal{L}$ is a set of $K$ tokens $\mathcal{L} = \{l_1, \ldots, l_K\}$, where a subset $\hat{\mathcal{L}} \subseteq \mathcal{L}$ of them are parametrized by a continuous parameter, i.e., each token $l \in \hat{\mathcal{L}}$ has an associated continuous parameter $\beta \in \mathcal{A}(l) \subset \mathbb{R}$, where $\mathcal{A}(l)$ is the token-dependent range. To ease the notation, we define $\bar{\mathcal{L}} := \mathcal{L} \setminus \hat{\mathcal{L}}$ and consider a dummy range $\mathcal{A}(l) = [0, 1] \subset \mathbb{R}$ for the strictly discrete tokens $l \in \bar{\mathcal{L}}$ and define,

$$l(\beta) = \begin{cases} l & \text{if } l \in \bar{\mathcal{L}} \\ l(\beta) & \text{if } l \in \hat{\mathcal{L}} \end{cases}, \forall (l, \beta) \in \mathcal{L} \times \mathcal{A}(l).$$

In other words, the parameter $\beta$ is ignored if $l \in \bar{\mathcal{L}}$. With this notation, we can write $\tau_i = l_i(\beta_i) \in \mathcal{L}, \forall i \in \{1, \ldots, T\}$. In the following, we use the notation $l_i(\beta_i) \equiv (l_i, \beta_i)$ and write

$$\tau = \langle \tau_1, \ldots, \tau_T \rangle = \langle l_1(\beta_1), \ldots, l_T(\beta_T) \rangle \equiv \langle (l_1, \beta_1), \ldots, (l_T, \beta_T) \rangle.$$

Given a sequence $\tau$, we define the *discrete skeleton* $\tau_d$ as the sequence obtained by removing the continuous parameters from $\tau$, i.e., $\tau_d = \langle (l_1, \cdot), \ldots, (l_T, \cdot) \rangle$. We introduce the operator $\texttt{eval} : \mathcal{T} \to \mathbb{T}$ to represent the semantic interpretation of the sequence $\tau$ as an object in the relevant design space $\mathbb{T}$.

The optimization problem is defined by the reward function $R : \mathbb{T} \to \mathbb{R}$, which can be deterministic or stochastic. In the stochastic case, we have a reward distribution $p_R(r|t)$ conditioned on the design $t \in \mathbb{T}$ and the reward function is given by $R(t) = \mathbb{E}_{r \sim p_R(r|t)}[r]$. Note that we do not assume that the reward function $R$ is differentiable with respect to the continuous parameters $\beta_i, \forall i$. In the following, we make a slight abuse of notation and use $R(\tau)$ and $p_R(r|\tau)$ to denote $R(\texttt{eval}(\tau))$ and $p_R(r|\texttt{eval}(\tau))$, respectively. The optimization problem is to find a sequence $\tau^* = \langle \tau_1^*, \ldots, \tau_T^* \rangle = \langle (l_1^*, \beta_1^*), \ldots, (l_T^*, \beta_T^*) \rangle$ such that $\tau^* \in \arg\max_{\tau \in \mathcal{T}} R(\tau)$.

We provide concrete examples of the concepts introduced above for the problems of symbolic regression (Section 4.2) and reinforcement learning with DT policies (Section 4.3).

### 3.2 METHOD

In applications of deep learning to combinatorial optimization (Bello et al., 2016), a probabilistic model $p(\tau)$ is learned over the configuration space $\mathcal{T}$. The model is trained to gradually allocate

probability mass to high scoring solutions. The training can be done using supervised learning, if problem instances with their corresponding solutions are available, or, more generally, using RL. In most cases, the model $p(\tau)$ is parameterized by a neural network (NN) with parameters $\theta$. More precisely, an autoregressive model is employed to generate a sequence $\tau$ in a step-by-step fashion. As the sequence unfolds, the model emits a vector of logits $\psi^{(i)}$ (for token $\tau_i$) conditioned on the previously generated tokens $\tau_{1:(i-1)}$, i.e., $\psi^{(i)} = \text{NN}(\tau_{1:(i-1)}, \theta)$.

Different model architectures can be employed to generate the logits $\psi^{(i)}$. For instance, recurrent neural networks (RNNs) have been utilized in (Petersen et al., 2021a; Landajuela et al., 2021c; Mundhenk et al., 2021; da Silva et al., 2023), and transformers with causal attention have been applied in works like (Biggio et al., 2021; Kamienny et al., 2022). The probability of selecting a token $\tau_i$ is given by the softmax of the logits $\psi^{(i)}$, i.e., $p(\tau_i|\tau_{1:(i-1)}, \theta) = \text{softmax}(\psi^{(i)})_{\mathcal{L}(\tau_i)}$, where $\mathcal{L}(\tau_i)$ is the index in $\mathcal{L}$ corresponding to node value $\tau_i$. Sequential token generation enables flexible configurations and the incorporation of constraints during the search process (Petersen et al., 2021a). Specifically, a prior $\psi_\circ^{(i)}$ is computed using all available information up to step $i-1$ and is added to the logits $\psi^{(i)}$ before sampling the token $\tau_i$.

Current deep learning approaches for combinatorial optimization only support discrete tokens, i.e., $\hat{\mathcal{L}} = \emptyset$, or completely decouple the discrete and continuous parts of the problem, by sampling first the discrete skeleton $\tau_d$ and then optimizing it offline (see Figure 1a). In this work, we extend these frameworks to support *joint optimization of discrete and continuous tokens*. The model is extended to emit two outputs $\psi^{(i)}$ and $\phi^{(i)}$ for each token $\tau_i = (l_i, \beta_i)$ conditioned on the previously generated tokens, i.e.,

$$\left(\psi^{(i)}, \phi^{(i)}\right) = \text{NN}((l, \beta)_{1:(i-1)}, \theta),$$

where we use the notation $(l, \beta)_{1:(i-1)}$ to denote the sequence of tokens $\langle(l_1, \beta_1), \ldots, (l_{i-1}, \beta_{i-1})\rangle$ (see Figure 1b). Given tokens $(l, \beta)_{1:(i-1)}$, the $i^{\text{th}}$ token $(l_i, \beta_i)$ is generated by sampling from the following distribution:

$$p((l_i, \beta_i)|(l, \beta)_{1:(i-1)}, \theta) = \begin{cases} \mathcal{U}_{[0,1]}(\beta_i)\text{softmax}(\psi^{(i)})_{\mathcal{L}(l_i)} & \text{if } l_i \in \bar{\mathcal{L}} \\ \mathcal{D}(\beta_i|l_i, \phi^{(i)})\text{softmax}(\psi^{(i)})_{\mathcal{L}(l_i)} & \text{if } l_i \in \hat{\mathcal{L}} \end{cases},$$

where $\mathcal{D}(\beta_i|l_i, \phi^{(i)})$ is the probability density function of the distribution $\mathcal{D}$ that is used to sample $\beta_i$ from $\phi^{(i)}$. Note that the choice of $\beta_i$ is conditioned on the choice of discrete token $l_i$. We assume that the support of $\mathcal{D}(\beta|l, \phi)$ is a subset of $\mathcal{A}(l)$ for all $l \in \hat{\mathcal{L}}$. Additional priors of the form $(\psi_\circ^{(i)}, 0)$ can be added to the logits before sampling the token $\tau_i$. See Appendix C for more details.

The parameters $\theta$ of the model are learned by maximizing the expected reward $J(\theta) = \mathbb{E}_{\tau \sim p(\tau|\theta)}[R(\tau)]$ or, alternatively, the quantile-conditioned expected reward

$$J_\varepsilon(\theta) = \mathbb{E}_{\tau \sim p(\tau|\theta)}[R(\tau)|R(\tau) \geq R_\varepsilon(\theta)],$$

where $R_\varepsilon(\theta)$ represents the $(1 - \varepsilon)$-quantile of the reward distribution $R(\tau)$ sampled from the trajectory distribution $p(\tau|\theta)$. It is worth noting that both objectives, $J(\theta)$ and $J_\varepsilon(\theta)$, serve as *relaxations* of the original optimization problem described above. Empirical evidence from Petersen et al. (2021a) demonstrates that, in practice, the $J_\varepsilon(\theta)$ objective tends to be more effective than $J(\theta)$ since it encourages the model to prioritize *best case* performance over *average case* performance.

To optimize the objective $J_\varepsilon(\theta)$, we use the risk-seeking policy gradient (Petersen et al., 2021a). In this case, the gradient of $J_\varepsilon(\theta)$ is given by

$$\nabla_\theta J_\varepsilon(\theta) = \mathbb{E}_{\tau \sim p(\tau|\theta)}\left[A(\tau, \varepsilon, \theta) \sum_{i=1}^{|\tau|} \begin{cases} \nabla_\theta \log p(l_i|(l, \beta)_{1:(i-1)}, \theta) & \text{if } l_i \in \bar{\mathcal{L}} \\ \nabla_\theta \log p(l_i|(l, \beta)_{1:(i-1)}, \theta)+ \\ \quad \nabla_\theta \log p(\beta_i|l_{1:i}, \beta_{1:i-1}, \theta) & \text{if } l_i \in \hat{\mathcal{L}} \end{cases}\bigg|A(\tau, \varepsilon, \theta) > 0\right],$$

where $A(\tau, \varepsilon, \theta) = R(\tau) - R_\varepsilon(\theta)$. See Appendix A for details about the derivation of the gradient and the practical implementation of the risk-seeking policy gradient. As in Petersen et al. (2021a), we add entropy to the loss function as a bonus. Since there is a continuous component in the library, the entropy for the distribution of $\tau_i$ in the sequence (see Appendix A) is

$$\mathcal{H}_i = \sum_{l \in \hat{\mathcal{L}}} p(l|(l, \beta)_{1:(i-1)}, \theta)\mathcal{H}_{\beta \sim \mathcal{D}(\beta|l, \phi_l)}(\beta|l) + \mathcal{H}_{l \sim p(l|(l, \beta)_{1:(i-1)}, \theta)}(l).$$

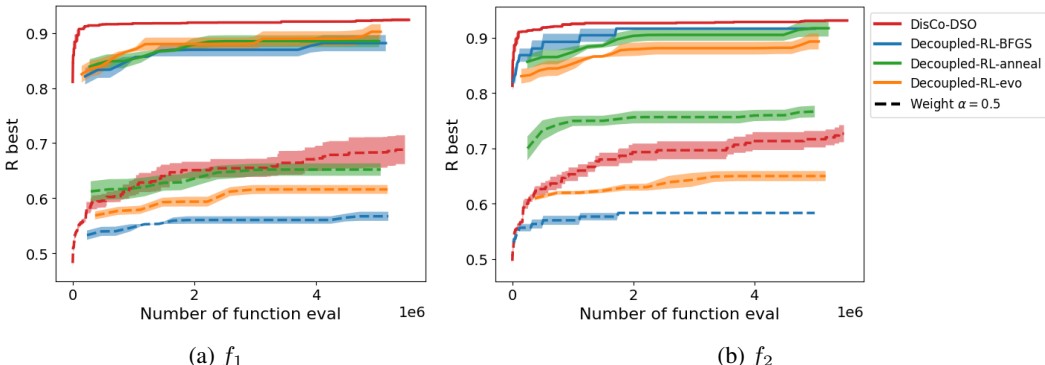

(a) $f_1$               (b) $f_2$

Figure 2: Reward of best solution versus number of function evaluations on a parameterized bitstring task, for two continuous optimization landscapes $f_1$ and $f_2$ and weights $\alpha = 0.5, 0.9$. Solid line corresponds to weight $\alpha = 0.9$, dashed line $\alpha = 0.5$. Mean and standard error over 5 seeds.

# 4 EXPERIMENTS

We demonstrate the benefits and generality of our approach on a diverse set of tasks as follows. Firstly, we introduce a new pedagogical task, called *Parameterized Bitstring*, to understand the conditions under which the benefits of DisCo-DSO versus decoupled approaches become apparent. We then consider the task of symbolic regression to show the benefits of DisCo-DSO on a more complex problem. Finally, we consider the task of DT policy optimization for interpretable reinforcement learning in standard control benchmarks.

**Baselines.** To demonstrate the advantages of joint discrete-continuous optimization, we compare our approach with the following classes of methods:

- Decoupled-RL-{BFGS, anneal, evo}: This baseline trains a generative model with reinforcement learning to produce a discrete skeleton, which is then optimized by a downstream nonlinear solver for the continuous parameters. The objective value at the optimized solution is the reward, which is used to update the generative model using the same policy gradient approach and architecture as DisCo-DSO. The continuous optimizer is either L-BFGS-B, simulated annealing (anneal) (Xiang et al., 1997), or differential evolution (evo) (Storn & Price, 1997), using the SciPy implementation (Virtanen et al., 2020).
- Decoupled-GP-{BFGS, anneal, evo}: This baseline uses genetic programming (GP) to produce a discrete skeleton, which is then optimized by a downstream nonlinear solver for the continuous parameters.

All experiments involving RL and DisCo-DSO use a RNN with a single hidden layer of 32 units as the generative model. The GP baselines use the Distributed Evolutionary Algorithms in Python software[1] (Fortin et al., 2012). Details on hyperparameters are provided in Appendix B.4.

## 4.1 PARAMETERIZED BITSTRING TASK

We design a general and flexible *Parameterized Bitstring* benchmark problem, denoted $\text{PB}(N, f, l^*, \beta^*)$, to test the hypothesis that DisCo-DSO is more efficient than the decoupled optimization approach. In each problem instance, the task is to recover a hidden string $l^* \in [0, 1]^N$ of $N$ bits and a vector of parameters $\beta^* \in \mathbb{R}^N$. Each bit $l_i^*$ is paired with a parameter $\beta_i^*$ via the reward function $R$, which gives a positive value based on an objective function $f(\beta_i, \beta_i^*) \in [0, 1]$ only if the correct bit $l_i^*$ is chosen at position $i$:

$$R(\tau, \beta) := \frac{1}{N} \sum_{i=1}^{N} \mathbb{1}_{\tau_i = \tau_i^*} \left( \alpha + (1 - \alpha) f(\beta_i, \beta_i^*) \right)$$

---

[1] https://github.com/DEAP/deap. LGPL-3.0 license.

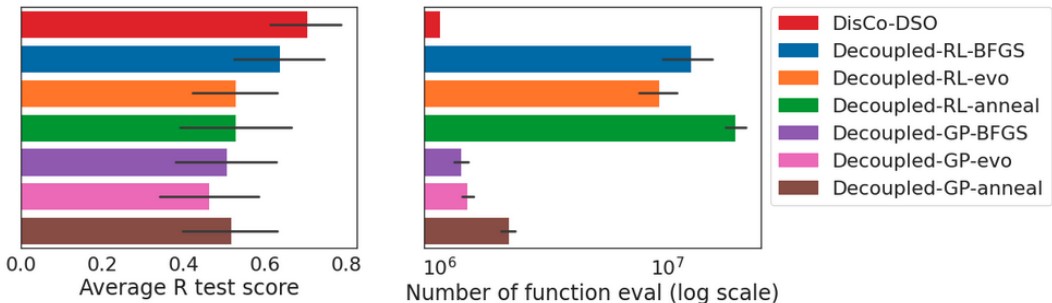

Figure 3: Visualizations of DisCo-DSO's reward on testing set and number of function evaluations used compared to other methods.

The scalar $\alpha \in [0,1]$ controls the relative importance of expending computational effort to optimize the discrete or continuous parts of the reward. The problem difficulty can be controlled by increasing the length $N$ and increasing the nonlinearity of the objective function $f$, such as by increasing the number of local optima. In our experiment, we tested the following objective functions, which represent objectives with multiple suboptimal local maxima ($f_1$) and discontinuous objective landscapes ($f_2$) (see Appendix B.1 for plots of these functions):

$$f_1(x, x^*) := \left| \frac{\sin(50(x - x^*))}{50(x - x^*)} \right|, \qquad f_2(x, x^*) := \begin{cases} 1, & |x - x^*| \leq 0.05 \\ 0.5, & 0.05 < |x - x^*| \leq 0.1 \\ 0, & 0.1 < |x - x^*| \end{cases} \qquad (1)$$

Figure 2 shows that DisCo-DSO is significantly more sample efficient than the decoupled approach when the discrete solution contributes more to the overall reward. This is because each sample generated by DisCo-DSO is a complete solution, which costs only one function evaluation to get a reward. In contrast, each sample generated by the baseline decoupled methods only has a discrete skeleton, which requires many function evaluations using the downstream optimizer to get a single complete solution. As the discrete skeleton increases in importance, the relative contribution of function evaluations for continuous optimization decreases.

## 4.2 SYMBOLIC REGRESSION FOR EQUATION DISCOVERY WITH CONSTANTS

Symbolic regression (SR) (Koza, 1994; Bongard & Lipson, 2007; Petersen et al., 2021a; Landajuela et al., 2021a) is a classical discrete-continuous optimization problem with applications in many fields, including robotics, control, and machine learning. In SR, we have $\mathcal{L} = \{x_1, \ldots, x_d, +, -, \times, \div, \sin, \cos, \ldots\}$ and $\hat{\mathcal{L}} = \{\text{const}(\beta)\}$, where $\text{const}(\beta)$ represents a constant with value $\beta$. The design space is a subset of the space of continuous functions, $\mathbb{T} \subset C(V^{\mathbb{R}})$, where $V \subset \mathbb{R}^d$ is the function support that depends on $\mathcal{L}$. The evaluation operator `eval` returns the function which expression tree has the sequence $\tau$ as pre-order traversal (depth-first and then left-to-right). For example, $\texttt{eval}(\langle +, \cos, y, \times, \text{const}(3.14), \sin, x \rangle) = \cos(y) + 3.14 \times \sin(x)$. Given a dataset $D = \{(x_1^{(i)}, \ldots, x_d^{(i)}, y^{(i)})\}_{i=1}^N$, the reward function is defined as the inverse of the normalized mean squared error (NMSE) between $y^{(i)}$ and $\texttt{eval}(\tau)(x_1^{(i)}, \ldots, x_d^{(i)}), \forall i \in \{1, \ldots, N\}$, computed as $\frac{1}{1+\text{NMSE}}$. SR has been shown to be NP-hard even for low-dimensional data (Virgolin & Pissis, 2022).

A key evaluation metric for symbolic regression is the *parsimony* of the discovered equations, i.e., the balance between the complexity of the identified equations and their ability to fit the data. A natural way to measure it is to consider the generalization performance over a test set: a SR method could find symbolic expressions that overfit the training data (using for instance overly complex expressions), but those expressions will not generalize well to unseen data.

For evaluating the generalization performance of various baselines, we rely on the benchmark datasets detailed in Table 1 of Appendix B.2. The test set is obtained by expanding the benchmark's domain $(a, b)$ and increasing the number of data points on which an expression is evaluated.

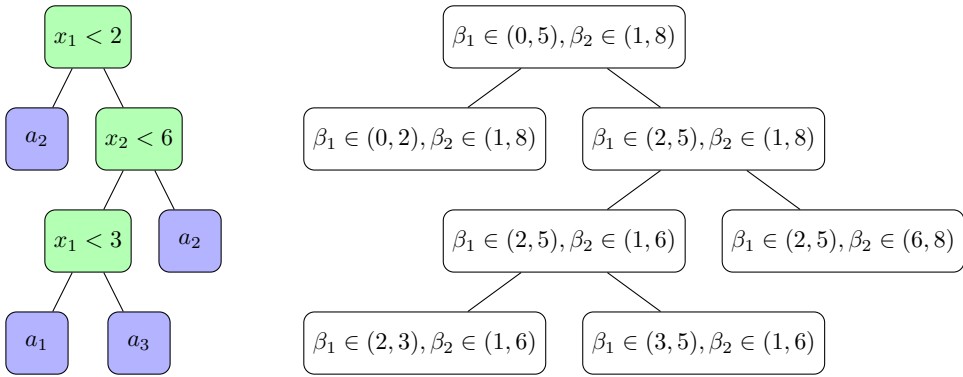

Figure 4: Left: the DT associated with the traversal $\langle x_1 < 2, a_2, x_2 < 6, x_1 < 3, a_1, a_3, a_2 \rangle$. Right: the corresponding bounds for the parameters during the sampling process (suppose the bounds for observations $x_1$ and $x_2$ are respectively [0, 5] and [1, 8]).

See Appendix B.2 for details. Since all experiments were conducted with 10 different random seeds, each random seed leads to a distinct best expression for that specific run. We take each of these 10 best expressions and compute the reward obtained on the evaluation dataset. Additionally, we calculate the number of function evaluations required to arrive at each expression. Subsequently, we aggregate the set of 10 evaluation rewards to calculate a single average reward and a single average number of function evaluations for each dataset. Following this procedure, we average over all the datasets to get an overall evaluation for each method.

Results in Figure 3 demonstrate the superior efficiency and generalization capability of DisCo-DSO in the SR setting. In particular, DisCo-DSO achieves the best average reward on the test set and the lowest number of function evaluations. Note that for DisCo-DSO we have perfect control over the number of function evaluations as it is determined by the number of samples ($10^6$ in this case). The Decoupled-GP methods exhibit a strong tendency to overfit to the training data and perform poorly on the test set. This phenomenon is known as the *bloat* problem in the SR literature (Silva & Costa, 2009). We observe that the joint optimization of DisCo-DSO is able to avoid this problem and achieve the best generalization performance.

### 4.3 DECISION TREE POLICIES FOR REINFORCEMENT LEARNING

In this section we consider the problem of discovering DT policies for RL. We consider $\mathbb{T}$ as the space of univariate DTs (Silva et al., 2020). Extensions to multivariate DTs, also known as oblique trees, are possible, but we leave them for future work. Given an environment with observations $x_1, \ldots, x_n$ and discrete actions $a_1, \ldots, a_m$, we consider the library of Boolean expressions and actions given by $\mathcal{L} = \{x_1 < \beta_1, \ldots, x_n < \beta_n, a_1, \ldots, a_m\}$, where $\beta_1, \ldots, \beta_n$ are the values of the observations that are used in the internal nodes of the DT. The evaluation operator $\texttt{eval} : \mathcal{T} \to \mathbb{T}$ is defined as follows. Given a sequence $\tau$ of nodes, we build the DT by traversing the nodes in $\tau$ from left to right. When traversing $\tau$, if a Boolean expression token ($x_n < \beta_n$) is encountered, the immediate next token is the left child of the node, and the next token after that is the right child of the node. If an action token ($a_n$) is encountered, the node is a leaf node. We use the convention that if a decision node is evaluated to true, the left child is taken, otherwise the right child is taken. See Figure 4 for an example. The reward function is defined as $R(t) = \mathbb{E}_{r \sim p_R(r|t)}[r]$ where $p_R(r|t)$ is the reward distribution following policy $t$ in the environment. In practice, we use the average reward over $N$ episodes, i.e, $R(t) = \frac{1}{N} \sum_{i=1}^{N} r_i$ where $r_i$ is the reward obtained in episode $i$ using policy $t$.

To efficiently sample decision nodes, we employ truncated normal distributions to select parameters $\beta_i$ within permissible ranges, which are determined by overall parameter constraints and the sampled parameter's value at the parent node. For instance, consider the DT displayed in Figure 4. Assume that the observation $x_1$ falls within the interval $[0, 5]$, and the tree commences with the node $x_1 < 2$. In the left child node, as $x_1 < 2$ is true, there is no need to evaluate whether $x_1$ is less than 4 (or any number between 2 and 5), as that is already guaranteed. Consequently, we should sample a parameter $\beta_1$ within the range $(0, 2)$. Simultaneously, since we do not assess the Boolean expression

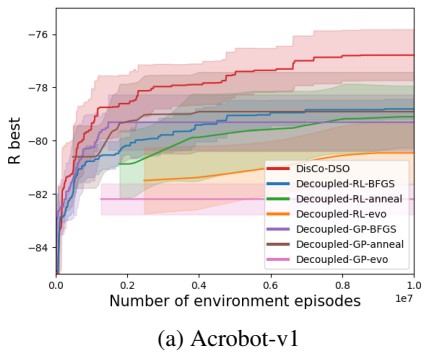
(a) Acrobot-v1

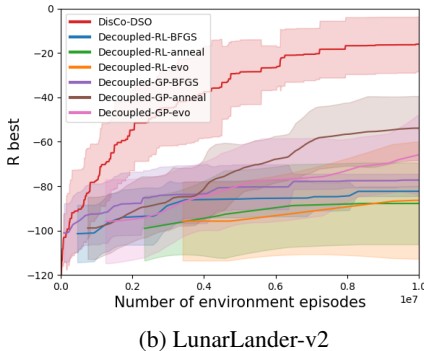
(b) LunarLander-v2

Figure 5: Reward of best solution versus number of function evaluations on the DT policy task, for Acrobot-v1 and LunarLander-v2.

regarding $x_2$, the bounds on $\beta_2$ remain consistent with those at the parent node. The parameter bounds for the remaining nodes are illustrated in Figure 4. The procedure for determining these maximum and minimum values is outlined in Algorithm 1 in Appendix C.

For evaluation, we follow other works in the field (Silva et al., 2020; Ding et al., 2020; Custode & Iacca, 2023) and use the OpenAI Gym (Brockman et al., 2016) environments MountainCar-v0, CartPole-v1, Acrobot-v1, and LunarLander-v2. We perform two different set of experiments for evaluation : sample-efficiency against baselines and best performance against literature.

In the first set of experiments, we investigate the sample-efficiency of DisCo-DSO on the DT policy task when compared to the baselines. In Figure 5 (see also Figure 10) in Appendix B.3), we report the best reward found so far versus number of environment episodes. We train each algorithm for 10 different random seeds and report the mean and standard deviation.

In the second set of experiments, we conduct a performance comparison of DisCo-DSO against various literature baselines, namely the evolutionary DTs as detailed in Custode & Iacca (2023), cascading DTs introduced in Ding et al. (2020), and interpretable differentiable DTs (DDTs) introduced in Silva et al. (2020). Whenever a method supplies a tree structure for a specific environment, we utilize the provided structure and assess it locally. In cases where the method's implementation is missing, we address this by leveraging open-source code. This approach allows us to train a tree on the absent environments, ensuring that we obtain a comprehensive result set for all evaluated methods across all environments. The DTs found by DisCo-DSO are shown in Figure 7 (see also Figure 11 in Appendix B.3). The comparisons are shown in Figure 6. Methods we trained locally are marked with an asterisk (*). In our evaluation process, all trees are rigorously assessed using a consistent set of 1,000 seeds on each environment, with one seed allocated per episode, resulting in a total of 1,000 episodes for each evaluation.

| | Acrobot-v1 | | CartPole-v1 | | LunarLander-v2 | | MountainCar-v0 | |
|---|---|---|---|---|---|---|---|---|
| **Algorithms** | MR | PC | MR | PC | MR | PC | MR | PC |
| DisCo-DSO | **-76.583** | 18 | **500.000** | 14 | **99.239** | 23 | **-100.966** | 15 |
| Evolutionary DTs | -97.115* | 5 | 499.582 | 5 | -87.617* | 17 | -104.933 | 13 |
| Cascading DTs | -82.136* | 58 | 496.628 | 22 | -227.019 | 29 | -200.000 | 10 |
| Interpretable DDTs | -497.863* | 15 | 389.786 | 11 | -120.376 | 19 | -172.209* | 15 |

Figure 6: Evaluation of the best univariate DTs found by DisCo-DSO and other baselines on the DT policy task. Here, MR is the mean reward earned in evaluation over a set of 1,000 random seeds, while PC represents the parameter count in each tree. For models trained in-house (*), the figures indicate the parameter count after the discretization process.

In Figure 6 we also show the complexity of the discovered DT as measured by the number of parameters in the tree. We count every (internal or leaf) node of univariate DTs (produced by all methods except for Cascading DTs) as one parameter. For Cascading DTs, the trees contain feature

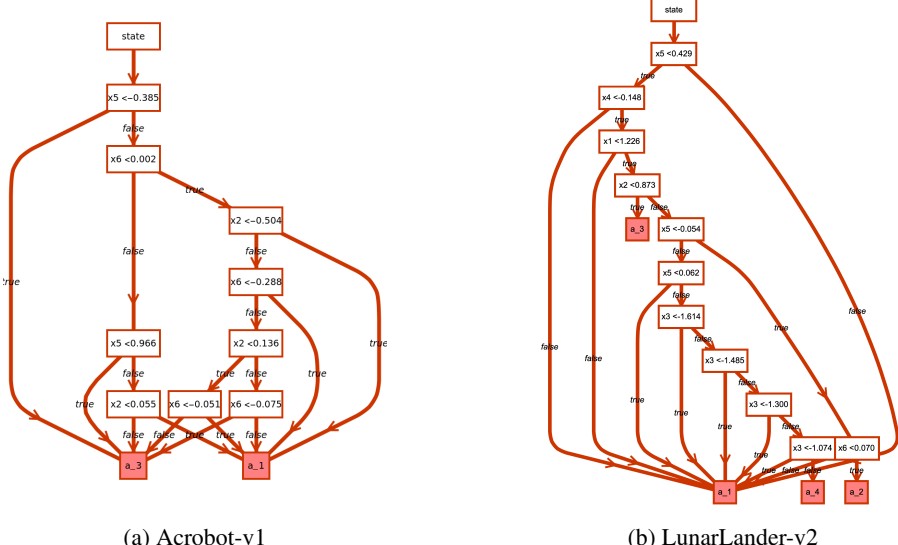

|  (a) Acrobot-v1 | (b) LunarLander-v2 |

Figure 7: Best DTs found by DisCo-DSO on the DT policy task for Acrobot-v1 and LunarLander-v2.

learning trees and decision making trees. The latter is just univariate DTs, so the same complexity measurement is used. For the leaf nodes of feature learning trees, the number of parameters is number of observations times number of intermediate features.

Figure 5 (and Figure 10 in Appendix B.3) show that DisCo-DSO dominates the baselines in terms of sample-efficiency. The trend is consistent across all environments, and more pronounced in the more complex environments. From Figure 6, we observe that the univariate DTs found by DisCo-DSO have the best performance on all environments at a comparable or lower complexity than the other literature baselines. The economical use of design evaluations by DisCo-DSO (each sample is a complete well-defined DT) versus the decoupled approaches, where each sample is a discrete skeleton that requires many evaluations to get a single complete solution, becomes a significant advantage in the RL environments where each evaluation involves running the environment for $N$ episodes.

## 5 CONCLUSION

In this work we have proposed DisCo-DSO (Discrete-Continuous Deep Symbolic Optimization), a novel approach for optimization in hybrid discrete-continuous spaces. DisCo-DSO uses a generative model to learn a joint distribution over discrete and continuous design variables to sample new hybrid designs. In contrast to standard decoupled approaches, in which the discrete skeleton is sampled first, and then the continuous variables are optimized separately, our joint optimization approach samples both discrete and continuous variables simultaneously. This leads to more efficient use of objective function evaluations, as the discrete and continuous dimensions of the design space can communicate with each other and guide the search. We have demonstrated the advantages of DisCo-DSO in tackling challenging problems such as symbolic regression and decision tree optimization. Notably, DisCo-DSO surpasses the current state-of-the-art techniques in the realm of univariate decision tree policy optimization for reinforcement learning.

As for the limitations of DisCo-DSO, it is important to highlight that the method relies on domain-specific information to define the ranges of continuous variables. In cases where this information is unavailable and estimations are necessary, the performance of DisCo-DSO could potentially be impacted. Furthermore, in our RL experiments, we constrain the search space to univariate decision trees. Exploring more complex search spaces, such as multivariate or oblique decision trees, remains an avenue for future research.

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

## A  ADDITIONAL ALGORITHM DETAILS

**Risk-seeking policy gradient for hybrid discrete-continuous action space.** The derivation of the risk-seeking policy gradient for the hybrid discrete-continuous action space follows closely the derivation in Petersen et al. (2021a) (see also Tamar et al. (2014)). The risk-seeking policy gradient for a univariate sequence $\tau$ is given by

$$\nabla_\theta J_\varepsilon(\theta) = \mathbb{E}_{\tau \sim p(\tau|\theta)} \left[ (R(\tau) - R_\varepsilon(\theta)) \nabla_\theta \log p(\tau|\theta) \mid R(\tau) \geq R_\varepsilon(\theta) \right].$$

In the hybrid discrete-continuous action space case, we have $\tau = \langle (l_1, \beta_1), \ldots, (l_T, \beta_T) \rangle$ and

$$p(\tau|\theta) = \prod_{i=1}^{|\tau|} p(\tau_i|\tau_{1:(i-1)}, \theta) = \prod_{i=1}^{|\tau|} p((l_i, \beta_i)|(l, \beta)_{1:(i-1)}, \theta). \tag{2}$$

Thus, using the convenient notation $A(\tau, \varepsilon, \theta) = R(\tau) - R_\varepsilon(\theta)$, the risk-seeking policy gradient for the hybrid discrete-continuous action space is given by

$$\nabla_\theta J_\varepsilon(\theta) = \mathbb{E}_{\tau \sim p(\tau|\theta)} \left[ A(\tau, \varepsilon, \theta) \nabla_\theta \log p(\tau|\theta) \mid A(\tau, \varepsilon, \theta) \geq 0 \right]$$

$$= \mathbb{E}_{\tau \sim p(\tau|\theta)} \left[ A(\tau, \varepsilon, \theta) \nabla_\theta \log \prod_{i=1}^{|\tau|} p((l_i, \beta_i)|(l, \beta)_{1:(i-1)}, \theta) \mid A(\tau, \varepsilon, \theta) \geq 0 \right]$$

$$= \mathbb{E}_{\tau \sim p(\tau|\theta)} \left[ A(\tau, \varepsilon, \theta) \sum_{i=1}^{|\tau|} \nabla_\theta \log p((l_i, \beta_i)|(l, \beta)_{1:(i-1)}, \theta) \mid A(\tau, \varepsilon, \theta) \geq 0 \right]$$

$$= \mathbb{E}_{\tau \sim p(\tau|\theta)} \left[ A(\tau, \varepsilon, \theta) \sum_{i=1}^{|\tau|} \left( \nabla_\theta \log p(l_i|(l, \beta)_{1:(i-1)}, \theta) + \nabla_\theta \log p(\beta_i|l_i, (l, \beta)_{1:(i-1)}, \theta) \right) \mid A(\tau, \varepsilon, \theta) \geq 0 \right]$$

$$= \mathbb{E}_{\tau \sim p(\tau|\theta)} \left[ A(\tau, \varepsilon, \theta) \sum_{i=1}^{|\tau|} \begin{cases} \nabla_\theta \log p(l_i|(l, \beta)_{1:(i-1)}, \theta), & \text{if } l_i \in \bar{\mathcal{L}} \\ \nabla_\theta \log p(\beta_i|l_i, (l, \beta)_{1:(i-1)}, \theta) + & \\ \quad \nabla_\theta \log p(l_i|(l, \beta)_{1:(i-1)}, \theta), & \text{if } l_i \in \hat{\mathcal{L}} \end{cases} \middle| A(\tau, \varepsilon, \theta) \geq 0 \right].$$

In practice, we use the following estimator for the risk-seeking policy gradient:

$$\nabla_\theta J_\varepsilon(\theta) \approx$$

$$\frac{1}{M} \sum_{i=1}^{M} \tilde{A}(\tau^{(i)}, \varepsilon, \theta) \sum_{j=1}^{|\tau^{(i)}|} \begin{cases} \nabla_\theta \log p(l_j^{(i)}|(l, \beta)_{1:(j-1)}^{(i)}, \theta), & \text{if } l_j^{(i)} \in \bar{\mathcal{L}} \\ \nabla_\theta \log p(\beta_j^{(i)}|l_j^{(i)}, (l, \beta)_{1:(j-1)}^{(i)}, \theta) + & \\ \quad \nabla_\theta \log p(l_j^{(i)}|(l, \beta)_{1:(j-1)}^{(i)}, \theta), & \text{if } l_j^{(i)} \in \hat{\mathcal{L}} \end{cases} \cdot \mathbb{I}(\tilde{A}(\tau^{(i)}, \varepsilon, \theta) \geq 0),$$

where $\tau^{(i)} = \langle (l_1^{(i)}, \beta_1^{(i)}), \ldots, (l_T^{(i)}, \beta_T^{(i)}) \rangle$ is the $i$-th trajectory sampled from $p(\tau|\theta)$, $M$ is the number of trajectories used in the estimator, and $A(\tau^{(i)}, \varepsilon, \theta) \approx \tilde{A}(\tau^{(i)}, \varepsilon, \theta) = R(\tau^{(i)}) - \tilde{R}_\varepsilon(\theta)$, with $\tilde{R}_\varepsilon(\theta)$ being an estimate of $R_\varepsilon(\theta)$.

**Entropy derivation in the hybrid discrete-continuous action space.** Recall that, for a distribution $\mathcal{D}$, the entropy is defined as

$$\mathcal{H}(\mathcal{D}) = - \int_{-\infty}^{\infty} \mathcal{D}(x) \log \mathcal{D}(x) \, dx.$$

In DisCo-DSO, we add an entropy regularization term $\mathcal{H}_i$ for each distribution $p((l_i, \beta_i)|(l, \beta)_{1:(i-1)}, \theta)$ encountered during the rollout. We have

$$\mathcal{H}_i = - \sum_{l_i \in \hat{\mathcal{L}}} \int_{-\infty}^{\infty} p((l_i, \beta_i)|(l, \beta)_{1:(i-1)}, \theta) \log p((l_i, \beta_i)|(l, \beta)_{1:(i-1)}, \theta) \, d\beta_i$$

$$- \sum_{l_i \in \bar{\mathcal{L}}} p(l_i|(l, \beta)_{1:(i-1)}, \theta) \log p(l_i|(l, \beta)_{1:(i-1)}, \theta)$$

$$= - \sum_{l_i \in \hat{\mathcal{L}}} \int_{-\infty}^{\infty} p(\beta_i|l_i) p(l) \log \left( p(\beta_i|l_i) p(l_i) \right) \, d\beta_i - \sum_{l_i \in \bar{\mathcal{L}}} p(l_i) \log p(l_i)$$

$$= - \sum_{l_i \in \hat{\mathcal{L}}} \int_{-\infty}^{\infty} p(\beta_i|l_i) p(l_i) \log p(\beta_i|l_i) \, d\beta_i - \sum_{l_i \in \hat{\mathcal{L}}} \int_{-\infty}^{\infty} p(\beta_i|l_i) p(l_i) \log p(l_i) \, d\beta_i - \sum_{l_i \in \bar{\mathcal{L}}} p(l_i) \log p(l_i)$$

$$= - \sum_{l_i \in \hat{\mathcal{L}}} p(l_i) \int_{-\infty}^{\infty} p(\beta_i|l_i) \log p(\beta_i|l_i) \, d\beta_i - \sum_{l_i \in \hat{\mathcal{L}}} p(l_i) \log p(l_i) \int_{-\infty}^{\infty} p(\beta_i|l_i) \, d\beta_i - \sum_{l_i \in \bar{\mathcal{L}}} p(l_i) \log p(l_i)$$

$$= \sum_{l_i \in \hat{\mathcal{L}}} p(l_i|(l, \beta)_{1:(i-1)}, \theta) \mathcal{H}_{\beta_i \sim \mathcal{D}(\beta_i|l_i, \phi_{l_i})}(\beta_i|l_i) - \sum_{l_i \in \mathcal{L}} p(l_i|(l, \beta)_{1:(i-1)}, \theta) \log p(l_i|(l, \beta)_{1:(i-1)}, \theta).$$

Note that we have removed the conditioning elements $\theta$ and $(l, \beta)_{1:(i-1)}$ in some terms in the above derivation for brevity.

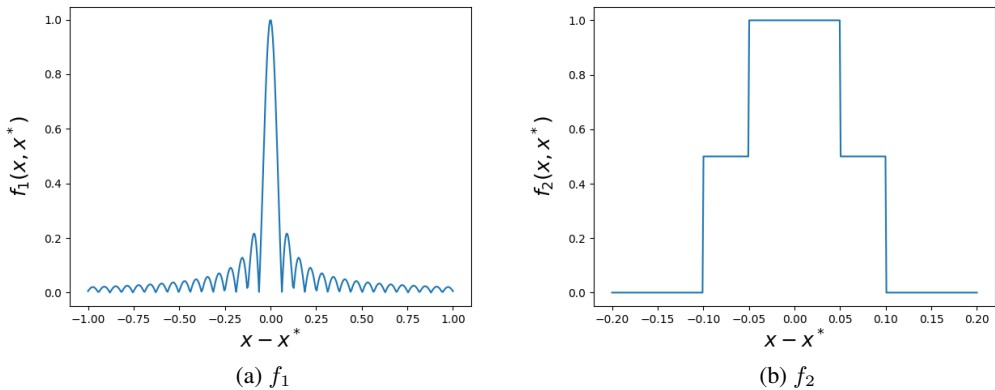

(a) $f_1$          (b) $f_2$

Figure 8: Objective functions in equation 1 against the difference $x - x^*$.

## B ADDITIONAL EXPERIMENTAL RESULTS

### B.1 PARAMETERIZED BITSTRING TASK

**Objective functions.** The objective functions $f_1$ and $f_2$ in Equation (1) are plotted in Figure 8. They are both non-differentiable and difficult to be optimized by Quasi-Newton methods.

### B.2 SYMBOLIC REGRESSION FOR EQUATION DISCOVERY WITH CONSTANTS

**Benchmarks.** In Table 1, we provide a compilation of benchmarks used in the symbolic regression task. The list comprises Livermore benchmarks from Mundhenk et al. (2021), Jin benchmarks from Jin et al. (2020), and Neat and Korn benchmarks from Trujillo et al. (2016). We also introduce Constant benchmarks, which are a variation of the Nguyen benchmarks from Uy et al. (2011) where floating constants are added to increase the complexity of the problem. Note that the selection of these benchmarks is not arbitrary. The linear and non-linear dependency of the expressions on $\beta_i$ is taken into consideration. Non-linear functions involving $\beta_i$ exhibit more complicated expressions, thus rendering more challenges for the optimization problem. On the other hand, linear functions, such as Jin-1, employ constants as coefficients for each variable term, thereby simplifying the optimization problems into linear regression problems. Consequently, we have selected approximately 25 non-linear functions and 20 linear functions, resulting in a total of 45 benchmark datasets. Exploring whether joint discrete-continuous optimization can outperform other classes of methods for both non-linear functions and linear functions can be a focus for future research.

**Evaluations sets.** To create the evaluation set, we adhere to a straightforward rule: we take the training set of each benchmark function, double the size of its domain, and double the number of points at which it is computed. For instance, a benchmark function with a training domain of $(-1, 1)$ and 20 points in that domain would have an evaluation set spanning $(-2, 2)$ with 40 data points within the expanded domain.

**Analysis of traversal lengths generated in Symbolic Regression.** Figure 9 shows the distribution of traversal lengths sampled by each method. These traversals are gathered from the top-performing expressions sampled, in the same way as the traversals used to generate Figure 3. Notice the extreme density placed at the maxmimum length (32) by the Decoupled-GP methods. This, paired with its poor generalization capability demonstrated in Figure 3a, leads us to conclude that the Decoupled-GP methods overfit heavily on the training data. The Decoupled-RL methods do this to a lesser degree, as does DisCo-DSO.

### B.3 DECISION TREE POLICIES FOR REINFORCEMENT LEARNING

**Results for MountainCar-v0 and CartPole-v1 environments.** In this section, we provide results for the MountainCar-v0 and CartPole-v1 environments. In Figure 10, we show the best reward

| Benchmark Name | Expression | Dataset |
|---|---|---|
| Livermore2-Vars2-2 | $x_1\,x_2\,\left(x_1 + x_1\,\sqrt{x_2\,(\beta_1 + x_1 + \beta_2\,x_1\,x_2 + \beta_3\,x_2^3)}\right)$ | U(-10,10,1000) |
| Livermore2-Vars2-4 | $\beta_1\,x_1 + \beta_2\,x_2 + (x_1 - x_2)^2$ | U(-10,10,1000) |
| Livermore2-Vars2-6 | $-\left((x_1 - x_1^2 - x_2)\,x_2\right) + \frac{(\beta_1\,x_1)}{(\beta_2 + \beta_3\,x_2)}$ | U(-10,10,1000) |
| Livermore2-Vars2-7 | $1 + \beta_1\,\sqrt{x_1} + \beta_2\,x_1 + \frac{(\beta_3\,x_2)}{x_1} + \beta_4\,\sqrt{-x_1 + x_2}$ | U(-10,10,1000) |
| Livermore2-Vars2-8 | $\beta_1\,x_1^2 + \beta_2\,x_2^2 + \beta_3\,x_2^3 + \frac{((\sqrt{x_1} - x_1)\,\log(x_2))}{\log(x_1)}$ | U(-10,10,1000) |
| Livermore2-Vars2-12 | $\sqrt{(x_1 + \frac{x_2}{e^{x_2}})}\,\log(\beta_1\,x_1^3 + \beta_2\,x_1^2\,x_2 + \beta_3\,x_2^3)$ | U(-10,10,1000) |
| Livermore2-Vars2-24 | $\beta_1 + \beta_2\,x_1 + \beta_3\,(x_1 - x_2^2)^2$ | U(-10,10,1000) |
| Livermore2-Vars3-4 | $\beta_1 + \beta_2 x_2 x_3 + x_1 + x_2 + \sqrt{\cos(x_2)} + x_3$ | U(-10,10,1000) |
| Livermore2-Vars2-9 | $\beta_1 + x_1 + x_1^2 - x_2 + \beta_2\,\sqrt{1 + \beta_3\,x_1 + \beta_4\,x_2}$ | U(-10,10,1000) |
| Livermore2-Vars2-16 | $-\left((x_1 - x_1^2 - x_2)\,x_2\right) + \frac{(\beta_1\,x_1)}{(\beta_2 + \beta_3\,x_2)}$ | U(-10,10,1000) |
| Livermore2-Vars2-17 | $x_1 - \sqrt{x_1}\,x_2 - \sin\left(\log(\beta_1\,x_1 + \beta_2\,x_1^3 + \beta_3\,x_2^2)\right)^2$ | U(-10,10,1000) |
| Livermore2-Vars2-19 | $\left(x_1 + \frac{(\beta_1 * \cos(x1))}{\sqrt{1 + \beta_2\,x_1 + \beta_3\,x_1\,(x_2^2 + \log(x_2))}}\right)\,(x_1 + \sin(1))$ | U(-10,10,1000) |
| Livermore2-Vars2-22 | $\frac{\log\left(\beta_1\,(-1 + \beta_2\,x_1)^2 + \frac{(e^{x_2}\,x_1)}{\sqrt{\cos(\beta_3 * x_2)}}\right)}{\sqrt{x_1}}$ | U(-10,10,1000) |
| Livermore2-Vars2-23 | $x_1^2 + \beta_1\,\sqrt{1 + \frac{(\beta_2\,(\beta_3 - x_2)\,(\beta_4 + \log x_2))}{x_1}}$ | U(-10,10,1000) |
| Livermore2-Vars3-2 | $\frac{(\beta_1\,(\beta_2\,x_1 - x_2)^2\,(\beta_4 - x_3)^2)}{(1 + \beta_3\,x_2^2)^2}$ | U(-10,10,1000) |
| Livermore2-Vars3-8 | $\frac{x_1}{\left((e^{(x_2 + x_2^4)} + \beta_1\,x_1^2 + \beta_2\,x_1^2\,x_3 + \beta_3\,x_3^2)\,(-x_2 + x_3 + \sqrt{\frac{\log(x_3)}{(x_1^2 + \sqrt{x_3})}})\right)}$ | U(-10,10,1000) |
| Livermore2-Vars3-9 | $x_1\,\sin\left(\frac{(\beta_1\,x_1)}{\sqrt{\beta_2\,x_1^2 - \frac{(x_1\,(-1 + x_1 + x_2^2)^4\,\sqrt{x_2 + \beta_3\,x_1\,x_3^2})}{x_3}}}\right)$ | U(-10,10,1000) |
| Livermore2-Vars3-11 | $\beta_1\,x_1 + \frac{x_1}{x_2} + \beta_2\,x_2 + \beta_3\,x_3$ | U(-10,10,1000) |
| Livermore2-Vars3-12 | $\beta_1 + \beta_2\,x_1 x_2^2 + x_2 + x_3$ | |
| Livermore2-Vars3-17 | $\beta_1\,x_1 + \beta_2\,x_2 + x_1\,\sqrt{x_2}\,\cos(x_1) + x_3 + 1$ | U(-10,10,1000) |
| Livermore2-Vars3-20 | $\beta_1 + \beta_2\,\sqrt{\frac{\sqrt{x_2}}{\sqrt{x_3}}} + x_1$ | U(-10,10,1000) |
| Livermore2-Vars3-24 | $\beta_1 + \beta_2\,x_1\,x_2^2 + x_2 + x_3$ | U(-10,10,1000) |
| Livermore2-Vars4-8 | $-x_1 + x_1\,\left(x_1 + x_4 + \sin\left(\frac{(-(e^{e^{x_3}}\,x_1) + x_2)}{(\beta_1\,x_1^2\,x_3 + \beta_2\,x_2^2\,x_3 + \beta_3\,x_3^3)}\right)\right)$ | U(-10,10,1000) |
| Livermore2-Vars4-16 | $x_3\,\left(\frac{\beta_1}{x_3} - x_4\right) + \frac{\beta_2\,x_4}{x_1} + \sqrt{x_2\,(x_1^2\,(-e^{x_2}) - x_2)}$ | U(-10,10,1000) |
| Livermore2-Vars4-18 | $x_1 + \sin\left(2\,x_2 + x_3 + \beta_1\,\sqrt{\beta_2 + \beta_3\,x_3^3 + x_2\,x_3^2} - e^{x_1}\,x_4 + \log((-x_1 + x_2)\,\log(x_2))\right)$ | U(-10,10,1000) |
| Livermore2-Vars4-23 | $-(\frac{x_1}{x_2}) + x_2 + \beta_1\,x_2\,x_3 + \frac{(\beta_2\,x_3)}{\sqrt{x_4}} + \beta_3\,\sqrt{x_4} + \log(x_1)$ | U(-10,10,1000) |
| Livermore2-Vars6-22 | $x_1 + \cos(x_2 + \beta_1\,x_5)\,(x_4 - \cos(x_6)\,\sin(\beta_2\,(\beta_3 + x_3 - x_4)))$ | U(-10,10,1000) |
| Livermore2-Vars6-23 | $x_1 + x_4 + \log\left(x_1^2 + x_1\,\left(\beta_1\,\sqrt{\beta_2\,x_1 + x_2} + \beta_3\,(x_3 - \frac{x_4}{x_5}) - x_6\right)\right)$ | U(-10,10,1000) |
| Livermore2-Vars6-24 | $\frac{\left(\beta_1\,\left(\beta_2\,\left(\beta_3\,x_2 + \frac{\sqrt{x_3}}{(\sqrt{-x_4 + x_5}\,x_6)}\right)^{1/4} + \sin(x_1)\right)\right)}{x_6}$ | U(-10,10,1000) |
| Livermore2-Vars7-14 | $\beta_1 x_6 + \beta_2\,x_1^2 x_6 - \cos\left(x_7\,\left(\frac{x_2^2\,x_5^2\,x_6^2\,(x_1 + x_3\,x_4\,x_6)^2 + x_6}{x_1} + x_7\right)\right)$ | U(-10,10,1000) |
| Livermore2-Vars7-23 | $-\left(\frac{(\sqrt{x_3}x_4)}{(x_3 + x_5 + \beta_1\,x_2\,x_3\,x_5 + \beta_2\,x_2\,x_6 + \beta_3\,x_4\,x_6\,x_7 + (-x_6 + x_7)^2)}\right) + x_1\,\cos(x_1) + \cos(x_2)$ | U(-10,10,1000) |
| Jin-1 | $\beta_1\,x_1^3 + \beta_2\,x_1^4 + \beta_3\,x_2 + \beta_4\,x_2^2$ | U(-3,3,100) |
| Jin-2 | $\beta_1 + \beta_2\,x_1^2 + \beta_3\,x_2^3$ | U(-3,3,100) |
| Jin-3 | $\beta_1\,x_1 + \beta_2\,x_1^3 + \beta_3\,x_2 + \beta_4\,x_2^3$ | U(-3,3,100) |
| Jin-6 | $\beta_1\,x_1\,x_2 + \beta_2\,\sin((\beta_3 + x_1)(\beta_4 + x_2))$ | U(-3,3,100) |
| Korn-12 | $2 + \beta_1\,\cos(\beta_2\,x_1)\,\sin(\beta_3\,x_5)$ | U(-50,50,100) |
| Neat-7 | $2 + \beta_1\,\cos(\beta_2\,x_1)\,\sin(\beta_3\,x_2)$ | U(-50,50,10000) |
| Constant-1 | $\beta_1\,x_1 + \beta_2\,x_1^2 + \beta_3\,x_1^3$ | U(-1,1,20) |
| Constant-2 | $\beta_1 + \sin\left(x_1^2\right)\cos(x_1)$ | U(-1,1,20) |
| Constant-3 | $\cos(x_1\,x_2)\,\sin(\beta_2\,x_1)$ | U(0,1,20) |
| Constant-4 | $\beta_1\,x_1^{x_2}$ | U(0,1,20) |
| Constant-5 | $\beta_1\,\sqrt{x_1}$ | U(0,4,20) |
| Constant-6 | $x_1^{\beta_1}$ | U(0,4,20) |
| Constant-7 | $2\,\cos(x_2)\,\sin(\beta_1\,x_1)$ | U(0,1,20) |
| Constant-8 | $\log(\beta_1 + x_1) + \log\left(\beta_2 + x_1^2\right)$ | U(0,4,20) |

Table 1: List of benchmarks used for symbolic regression task. Each benchmark includes input variables denoted as $x_1, x_2, \ldots, x_d$, along with floating constants denoted as $\beta_i$. The notation $\mathrm{U}(a, b, c)$ denotes the inclusion of $c$ random points uniformly sampled from the open interval $(a, b)$ for each input variable $x_i$.

versus number of environment episodes for MountainCar-v0 and CartPole-v1. Figure 11 shows the best decision trees found by DisCo-DSO for MountainCar-v0 and CartPole-v1.

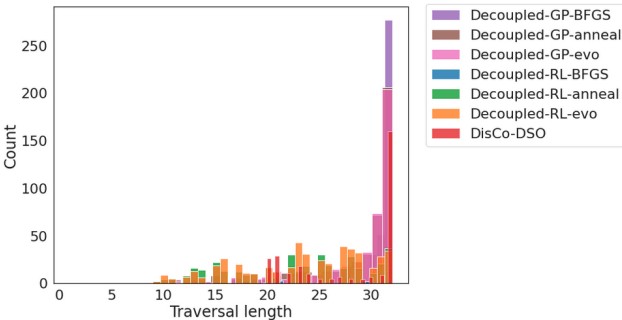

Figure 9: Histogram illustrating the lengths of traversals selected by different methods on the SR task.

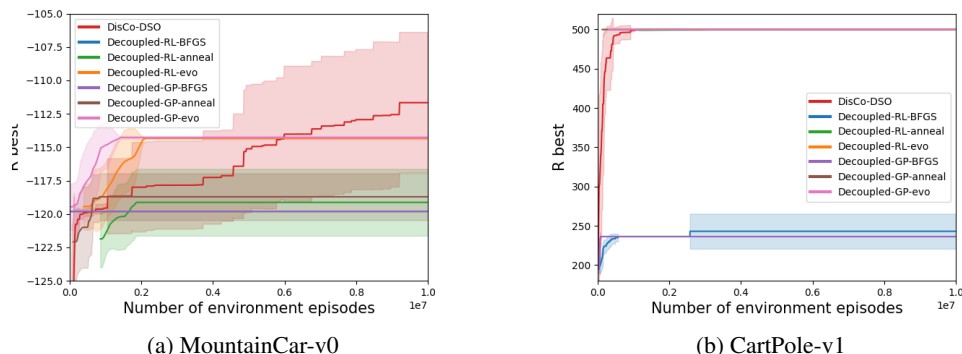

(a) MountainCar-v0                     (b) CartPole-v1

Figure 10: Reward of best solution versus number of function evaluations on the decision tree policy task for MountainCar-v0 and CartPole-v1.

**Note on oblique decision trees.** Custode & Iacca (2023) consider multivariate, or oblique decision trees. These are trees where the left-hand side of a decision node is composed of an expression, while the right-hand side is still a boolean decision parameter $\beta_n$. While these trees perform well on more complex environments such as LunarLander-v2 (published results report an average test score of 213.09), we do not compare against them here as the search space is drastically different.

### B.4 HYPERPARAMETERS

In Table 2, we provide the common hyperparameters used for the RL-based generative methods (DisCo-DSO and Decoupled-RL). The hyperparameters for the GP-based method are provided in Table 3. DisCo-DSO's specific hyperparameters, linked to modeling of the distribution $\mathcal{D}$, are provided in Table 4. For the DT policies for reinforcement learning task, the parameter $N$ (number of episodes to average over to compute a single reward $R(\tau)$) is set to 100.

## C  ADDITIONAL CONSTRAINTS FOR DECISION TREES

The autoregressive sampling used by DisCo-DSO allows for the incorporation of user-specified priors and constraints. These priors and constraints are applied *in situ*, i.e., during the sampling process. These ideas have been used by several works using similar autoregressive sampling procedures (Popova et al., 2019; Petersen et al., 2021a;b; Landajuela et al., 2021b; Mundhenk et al., 2021; Kim et al., 2021). In this work, we include three additional constraints, one on the continuous parameters, two on the discrete tokens.

**Constraint on parameter range.** As noted in Section 4.3, by using the truncated normal distribution, upper/lower bounds are imposed on the parameters of decision trees (i.e., $\beta_n$ in the Boolean

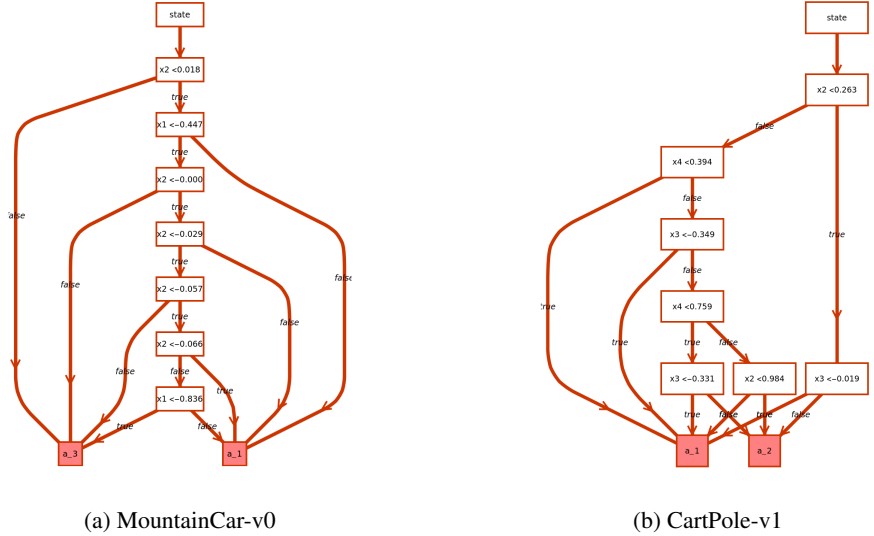

(a) MountainCar-v0     (b) CartPole-v1

Figure 11: Best decision trees found by DisCo-DSO on the decision tree policy task for MountainCar-v0 and CartPole-v1.

| Parameter | Value |
|---|---|
| Optimizer | Adam (Kingma & Ba, 2017) |
| Number of layers | 1 |
| Number of hidden units | 32 |
| RNN type | LSTM (Hochreiter & Schmidhuber, 1997) |
| Learning rate ($\alpha$) | 0.001 |
| Entropy coefficient ($\lambda_{\mathcal{H}}$) | 0.01 |
| Moving average coefficient ($\beta$) | 0.5 |
| Risk factor ($\epsilon$) | 0.2 |

Table 2: DisCo-DSO and Decoupled-RL hyperparameters

| Parameter | Value |
|---|---|
| Population size | 1,000 |
| Generations | 1,000 |
| Fitness function | NRMSE |
| Initialization method | Full |
| Selection type | Tournament |
| Tournament size ($k$) | 5 |
| Crossover probability | 0.5 |
| Mutation probability | 0.5 |
| Minimum subtree depth ($d_{\min}$) | 0 |
| Maximum subtree depth ($d_{\max}$) | 2 |

Table 3: Decoupled-GP hyperparameters

| Parameter | Value |
|---|---|
| Parameter shift | 0.0 |
| Parameter generating distribution scale ($\sigma$) | 0.5 |
| Learn parameter generating distribution scale | False |
| Parameter generating distribution type | Normal |

Table 4: DisCo-DSO specific hyperparameters

---

**Algorithm 1** Finding bounds for parameters in decision trees

---

1: **input** parent token $l_p(\beta_p)$, bounds of the parameters of the parent token $\beta_p^{\max}, \beta_p^{\min}$
2: **hyperparameters** resolution $h > 0$
3: **output** bounds for the parameters of the next token $\beta_i^{\max}, \beta_i^{\min}$
4: $\beta_i^{\max}, \beta_i^{\min} \leftarrow \beta_p^{\max}, \beta_p^{\min}$          ▷ Inherit parameter bounds from parent
5: **if** the next token is a right child of $l_p(\beta_p)$ **then**
6:      $(\beta_i^{\min})_{l_p} \leftarrow \beta_p + h$          ▷ Adjust the lower bound corresponding to $l_p$
7:      **if** $(\beta_i^{\max})_{l_p} - (\beta_i^{\min})_{l_p} < h$ **then**
8:          $(\beta_i^{\min})_{l_p} \leftarrow (\beta_i^{\max})_{l_p} - h/2$          ▷ Maintain a minimal distance between bounds
9:      **end if**
10: **else**
11:      $(\beta_i^{\max})_{l_p} \leftarrow \beta_p - h$          ▷ Adjust the upper bound corresponding to $l_p$
12:      **if** $(\beta_i^{\max})_{l_p} - (\beta_i^{\min})_{l_p} < h$ **then**
13:          $(\beta_i^{\max})_{l_p} \leftarrow (\beta_i^{\min})_{l_p} + h/2$          ▷ Maintain a minimal distance between bounds
14:      **end if**
15: **end if**
16: **return** $\beta_i^{\max}, \beta_i^{\min}$

---

expression tokens $x_n < \beta_n$) to prevent meaningless internal nodes from being sampled. In Algorithm 1, we provide the detailed procedure for determining the upper/lower bounds at each position of the traversal. The resolution $h > 0$ is a hyperparameter that controls the distance between the parameters $\beta_p$ at the parent node and the corresponding bounds at the children nodes. This guarantees that the sampled decision trees must have finite depth if the environment-enforced bounds on the features of the optimization problem are finite. Moreover, it also prevents the upper/lower bounds from being too close, which can lead to numerical instability in the truncated normal distribution.

**Constraint on Boolean expression tokens.** Depending on the values of the parameter bounds, we also impose constraints on the discrete tokens $x_n < \beta_n$. Specifically, when the upper/lower bounds $\beta_n^{\max}$ and $\beta_n^{\min}$ for the parameter of the $n$-th Boolean expression token $x_n < \beta_n$ are too close, oftentimes there is not much value to split the $n$-th feature space further. Therefore, if $\beta_n^{\max} - \beta_n^{\min} < h$, where $h$ is the resolution hyperparameter in Algorithm 1, then $x_n < \beta_n$ are constrained from being sampled.

**Constraint on discrete action tokens.** If the left child and right child of a Boolean expression token $x_n < \beta_n$ are the same discrete action token $a_j$, the subtree will just be equivalent to a single leaf node containing $a_j$. We add a constraint that if the left child of $x_n < \beta_n$ is $a_j$, then the right child cannot be $a_j$.

