# OpenReview forum: "DisCo-DSO: Coupling Discrete and Continuous Optimization for Efficient Generative Design in Hybrid Spaces"
_ICLR.cc/2024/Conference — Submitted to ICLR 2024_

### Official Review · Reviewer_gpUA · 2023-10-29

**Soundness:** 2 fair
**Presentation:** 2 fair
**Contribution:** 2 fair
**Rating:** 5
**Confidence:** 3

**Summary:**

DisCo-DSO is a novel approach for optimizing in hybrid discrete-continuous spaces. It uses a generative model to jointly optimize discrete and continuous variables, leading to improved performance and efficiency, especially in complex optimization tasks like interpretable reinforcement learning with decision trees.

**Strengths:**

I believe that research focusing on effectively exploring solutions while considering the impact of discrete and continuous variables on the objective function has its significance. This study proposes research that optimizes both discrete and continuous variables simultaneously, extending the conventional method of constructing solutions used in AI-based combinatorial optimization research to a continuous approach.

**Weaknesses:**

I believe there are various approaches to solving optimization problems that involve a mix of continuous and discrete variables. Instead of merely extending the existing modeling structure, I don't consider optimizing both discrete and continuous variables simultaneously as a significant contribution in itself. Research that effectively explores solutions while considering the impact of discrete and continuous variables on the objective function holds its own merit. This study, by proposing optimization of both discrete and continuous variables simultaneously and expanding the traditional approach used in AI-based combinatorial optimization research to a continuous one, may need to offer more than just an extension of the existing modeling structure to make a substantial contribution.

**Questions:**

1. I'm having difficulty understanding why an autoregressive policy structure is necessary for generating solutions to optimization problems involving a mix of continuous and discrete variables. While sequential structures are commonly used in reinforcement learning for combinatorial optimization problems to effectively learn policies for arbitrary problems, this study appears to be focusing on optimizing a specific given problem. In such a case, wouldn't it be more efficient to explore the entire solution space rather than constructing solutions sequentially?

2. How is the order for selecting optimization variables determined, and does this order have the potential to affect the performance?

3. Mixed Integer Programming (MIP) is a well-known class of optimization problems that involve optimizing both discrete and continuous variables, and many studies attempt to optimize MIPs using deep learning. How does this research differ from those studies involving MIPs?

4. The experimental content seems limited. While three types of problems are presented, the first problem only compares the number of function evaluations, and the second problem presents curves indicating the improvement in solutions for a specific problem. It would be beneficial to solve a more diverse set of problems and provide statistical evidence to validate the performance.

---

> ### Author Response · Authors · 2023-11-14
> **Rebuttal by Authors to Reviewer gpUA**
>
> We thank the reviewer for her/his comments and suggestions. We will address the reviewer's comments in the following.
>
> **1. "Why is an autoregressive policy structure is necessary? ... this study appears to be focusing on optimizing a specific given problem. In such a case, wouldn't it be more efficient to explore the entire solution space rather than constructing solutions sequentially?"**
> We would like to emphasize that DisCo-DSO is explicitly tailored for generative optimization scenarios where the solution is constructed sequentially.
> In this context, the sequential construction of the solution arises due to the following factors:
> 1. The length of the sequence is variable.
> 2. The order of the sequence is not predetermined but is determined by the model.
> 3. There exist prefix-dependent positional constraints (or priors) dictated by the specific problem at hand. For instance, in symbolic regression, the model should not be allowed to add a trigonometric operator if it has already done so in the past to avoid nested trigonometric operators.
>
> The reviewer is right in that we are solving specific problem instances, but even then, the above factors are present. An autoregressive policy is well-suited to address these factors. We will clarify this in the camera-ready version of the paper.
>
> **2. "How is the order for selecting optimization variables determined, and does this order have the potential to affect the performance?"**
> In this work, we are doing *de novo* design: we start from an empty set and let the model decide which and how many tokens
> (or building blocks) to add to the solution. The length of the solution is variable and the order in which the tokens are added is determined by the model. The ordering is a decision variable of the optimization problem (we can not fix it a priori). The reviewer is right that, in the tasks of symbolic regression and decision tree search, the order in which the tokens are added will affect the final design and thus the performance. The model should learn to add the tokens in the order that maximizes the reward. We will clarify this in the camera-ready version of the paper.
>
> **3. "How does this research differ from those studies involving MIPs?"**
> The main differences between DisCo-DSO and approaches that use deep learning to solve MIP problems are the following:
> 1. The search space in DisCo-DSO is less structured than in MIP problems. The solutions in DisCo-DSO are variable length and the order in which the tokens are added is determined by the model. In MIP problems, the solutions are fixed length and the order is fixed.
> 2. In DisCo-DSO we have positional constraints that depend on the prefix of the solution. For instance, in symbolic regression, the model should not be allowed to add a trigonometric operator if it has already done so in the past to avoid nested trigonometric operators. In general MIP problems, the constraints are fixed and do not depend on the prefix of the solution.
> 3. In DisCo-DSO, discrete and continuous variables are linked. For instance, in decision tree search, we have the discrete variable $x_i < \beta$ and the continuous variable $\beta =5.3$, that jointly form the decision tree node $x_i < 5.3$. The search mechanism is designed to optimize both variables jointly. It is unclear if the approaches that use deep learning to solve MIP problems can handle this type of coupling. We will clarify this in the camera-ready version of the paper.
>
> We will add a discussion about the differences between DisCo-DSO and approaches that use deep learning to solve MIP problems in the camera-ready version of the paper.
>
> **4. "It would be beneficial to solve a more diverse set of problems and provide statistical evidence to validate the performance".**
> In this paper, we wanted to investigate the benefits of DisCo-DSO over the standard decoupled approach in generative optimization scenarios where the solution is constructed sequentially.
> For generality, we wanted to cover deterministic and stochastic optimization problems and also, different search spaces and semantics. We chose
> symbolic regression (deterministic optimization over the space of mathematical expressions) and decision tree policies in RL (stochastic optimization over the space of decision trees) as representative examples of these scenarios.
> The implementation of each of these tasks in non-trivial and required a significant amount of work. We believe that the results are conclusive and statistically significant, and show that DisCo-DSO outperforms the standard decoupled approach in terms of generalization over the test data and quality of solution per number of function evaluations.

---

> > ### Comment · Reviewer_gpUA · 2023-11-23
> > **Thank you for the response.**
> >
> > Thank you for providing clarification through the rebuttal; I have raised my score to 5. However, I still believe that the novelty and scalability of this research are lacking.

---

### Official Review · Reviewer_6T4T · 2023-10-31

**Soundness:** 3 good
**Presentation:** 4 excellent
**Contribution:** 3 good
**Rating:** 6
**Confidence:** 3

**Summary:**

The authors present a method for using reinforcement learning to train a model that outputs a sequence of both discrete and corresponding continuous values. The goal here is to tackle problems that involve predicting an object that has both discrete and continuous values such as a decision tree with selection of branching feature as well as the corresponding threshold, or producing a symbolic regression expression which is a combination of selected functions and constants. Their approach jointly produces both discrete symbol and a corresponding continuous value as opposed to previous work which focused on generating discrete backbones and then found the continuous values for the fixed skeleton. They evaluate their approach empirically by comparing against baselines that handle discrete and continuous decisions independently, as well as baselines from the literature for symbolic regression and interpretable RL (identifying a decision tree for solving simple RL problems). They demonstrate improved performance over previous approaches both in terms of solution quality, as well as in the efficiency as they require fewer calls to the evaluation metric to train their approach since the continuous variables don’t need to be optimized separately.

**Strengths:**

The main strength of the proposed approach is the empirical performance seems to be much better than previous approaches for solving optimization over hybrid discrete and continuous spaces. The results seem to indicate substantial improvement from the given model, and the evaluation is done on both symbolic regression and interpretable RL which are quite diverse domains. Furthermore, the approach itself may have broader impact for other settings in hybrid design spaces, and in identifying interpretable machine learning models.

**Weaknesses:**

Some of the small weaknesses here are in the motivation of the approach as well as tackling problems which require solutions that are more heavily constrained.
The approach is partially motivated by the idea that jointly generating the discrete and continuous objects makes the reward more aligned with the solution itself rather than approaches which generate the discrete backbone and then optimize the continuous variables after the fact. However, it seems that this approach also may have misleading rewards for the skeleton or continuous solution if neither are optimal. It may shy away from high quality discrete solutions even though there may be one setting of continuous variables which are highly performant.
It is also unclear how this approach may work on more complicated discrete-continuous settings where the feasible region may be more complex such as in solving mixed integer linear programming, or in other cases where the continuous space may be more decoupled from the discrete space such as in cases where there are many continuous decisions that need to be made but not all of them have a corresponding discrete decision.
Another small limitation is that it would be helpful to see the applicability of this approach on more tasks that have hybrid domains such as real world use cases of symbolic regression.

**Questions:**

How does this method avoid the issue of poor or misleading rewards for the right discrete skeleton? It might be the case that the continuous predictions are incorrect while the skeleton is correct.

Is it possible to generate continuous decisions that are unrelated to discrete decisions and thus uncoupled?

---

> ### Author Response · Authors · 2023-11-14
> **Rebuttal by Authors to Reviewer 6T4T**
>
> We highly appreciate the reviewer's comments and suggestions. Her or his comments show that the reviewer has read the paper carefully and understood the main contributions of the paper. We will address the reviewer's comments in the following.
>
> **"However, it seems that this approach also may have misleading rewards for the skeleton or continuous solution if neither are optimal. It may shy away from high quality discrete solutions even though there may be one setting of continuous variables which are highly performant"**
> Our results in symbolic regression and decision tree search do not suggest that the model is shying away from high quality discrete solutions. On the contrary, the results show that a coupled approach outperforms a decoupled approach in terms of generalization over the test data and quality of solution per number of function evaluations.
>
> The reviewer is right that there could be a situation where DisCo-DSO produce the optimal discrete skeleton and fail to find the optimal continuous solution (this could also happen in a decoupled approach). However, the opposite is often true in decoupled approaches: we find a poor discrete skeleton and we make it work by overfitting the continuous parameters to the data. This can have the effect of confusing the model and lead to a suboptimal solution. As we show in the experiments in the paper, this effect is catastrophic in terms of generalization over the test data (parsimony of the produced solutions) and wasted function evaluations.
>
> **"It is also unclear how this approach may work on more complicated discrete-continuous settings where the feasible region may be more complex such as in solving mixed integer linear programming"**
> We would like to emphasize that DisCo-DSO is explicitly tailored for generative modeling scenarios where :
> 1. The length of the solution is variable.
> 2. The order of the solution is not predetermined but is determined by the model.
> 3. There exist prefix-dependent positional constraints (or priors) dictated by the specific problem at hand.
> Standard mixed integer linear programming problems problems do not satisfy these conditions.
>
> That said, we believe that DisCo-DSO can be extended to mixed integer linear programming problems using the $\text{const}(\beta)$ operator (similar to symbolic regression). We could then use rejection sampling to sample from the feasible region. Although interesting, we believe that this is beyond the scope of this paper. We will clarify this in the camera-ready version of the paper.
>
> **"...or in other cases where the continuous space may be more decoupled from the discrete space such as in cases where there are many continuous decisions that need to be made but not all of them have a corresponding discrete decision".**
> DisCo-DSO would support this case by using the $\text{const}(\beta)$ operator (similar to symbolic regression). Note that $\beta$ does not need to be related to a discrete decision. The relation is only established by the ${\tt eval}$ operator. For instance, consider the fully continuous problem of finding the maximum of a function $f(\boldsymbol{x})$ with $\boldsymbol{x} \in \mathbb{R}^n$. In this case, we will have
> $\mathcal{L}= \hat{\mathcal{L}}=\{\text{const}(\beta)\}$. DisCo-DSO will produce skeletons $\langle \text{const}(\beta_1), \dots, \text{const}(\beta_n) \rangle$ and the $\beta_i$'s will be stochastically optimized by standard expectation–maximization. We will clarify this in the camera-ready version of the paper.
>
> **Application to real world use cases of symbolic regression.**
> The objective of the experiments in the symbolic regression task was to investigate the sample efficiency of DisCo-DSO compared to traditional decoupled approaches. We believe that those conclusions will be valid regardless of the origin of the data.
>
> **"How does this method avoid the issue of poor or misleading rewards for the right discrete skeleton?"**
> Our empirical evaluation in symbolic regression and decision tree search suggests that DisCo-DSO does not suffer from this issue, but rather outperforms decoupled approaches in terms of generalization over the test data and quality of solution per number of function evaluations. See above for more details.
>
> **"Is it possible to generate continuous decisions that are unrelated to discrete decisions and thus uncoupled?"**
> We addressed this question above. The short answer is yes. We will clarify this in the camera-ready version of the paper.

---

> > ### Comment · Reviewer_6T4T · 2023-11-21
> >
> > I thank the authors for providing responses to my original comments. Additionally, the authors have provided more results demonstrating their performance improvements against baselines in their experimental settings.
> >
> > However, given that the main support for the claims in this paper is empirical, it would be good to give stronger empirical evidence that this method works in some real world setting. One way of doing this might be to provide results for symbolic regression in some symbolic regression dataset such as those proposed in [1], or the SRBench competition [2] that was used in the referenced previous work the authors build off of [3]. Given the demonstrated improvement in interpretable RL domains, it seems that this approach has potential to be generally applicable and I wouldn’t be against having it accepted. However, I believe that more could be done to strengthen the claims of empirical performance over previous approaches for the task of symbolic regression.
> >
> >
> > [1] La Cava, William, et al. "Contemporary Symbolic Regression Methods and their Relative Performance." Thirty-fifth Conference on Neural Information Processing Systems Datasets and Benchmarks Track (Round 1). 2021.
> >
> > [2] https://cavalab.org/srbench/competition-2022/#real-world-track-rankings
> >
> > [3] Brenden K. Petersen, Mikel Landajuela, T. Nathan Mundhenk, Cl´audio Prata Santiago, Sookyung Kim, and Joanne Taery Kim. Deep symbolic regression: Recovering mathematical expressions from data via risk-seeking policy gradients. In 9th International Conference on Learning Rep- resentations, ICLR 2021, Virtual Event, Austria, May 3-7, 2021.

---

### Official Review · Reviewer_CEut · 2023-11-01

**Soundness:** 3 good
**Presentation:** 3 good
**Contribution:** 2 fair
**Rating:** 5
**Confidence:** 3

**Summary:**

As opposed to de-coupling the discrete and continuous representations, the authors concatenate the two of these are use autoregressive methods for optimization. The approach is applied to a toy problem in symbolic regression and reinforcement learning for decision trees.

**Strengths:**

The combination of discrete and continuous representations is an important direction for research, with the neuro-symbolic community making a lot of progress and several application domains of interest, including symbolic regression, combinatorial optimization, and symbolic distillation. The pedagogical example is an intuitive way to demonstrate the utility of the approach.

**Weaknesses:**

The concatenation of discrete and continuous variables into a single vector and the higher-level approach are very straightforward and their novelty seems limited.

**Questions:**

Why are you comparing your method against the baselines you define as opposed to using baselines from the literature in the case of symbolic regression (Figure 3)? (For decision trees, baselines from the literature are used). Given recent work on neuro-symbolic regression, the authors should compare against methods in the literature.

---

> ### Author Response · Authors · 2023-11-18
> **Rebuttal by Authors to Reviewer CEut**
>
> We thank the reviewer for the comments and suggestions. We will address the reviewer's comments in the following.
>
> **"The concatenation of discrete and continuous variables into a single vector and the higher-level approach are very straightforward and their novelty seems limited."**
> We agree with the observation that the concatenation of discrete and continuous variables into a single vector, along with the higher-level approach, may seem straightforward. However, we would like to highlight that the novelty of our approach lies in its application to the specific problem of optimization with generative models for hybrid spaces.
> The integration involves several challenges that we address in the paper, such as the need for prefix-dependent positional constraints, and the extension of the gradient update to handle the new search space. To the best of our knowledge, this is the first instance where such a concept has been systematically employed in this context. We believe this application of the idea adds a significant contribution to the field and opens up new avenues for research.
>
> **"Why are you comparing your method against the baselines you define as opposed to using baselines from the literature in the case of symbolic regression (Figure 3)?"**
> The paper already contains well-established baselines for symbolic regression. In particular:
>
> 1. "Decoupled-GP-BFGS": This baseline corresponds to a standard implementation of Genetic Programming for symbolic regression à la Koza, 1994. This represents the current most common approach to symbolic regression in the literature.
> 2. "Decoupled-RL-BFGS": This baseline corresponds exactly to the method "Deep Symbolic Regression" from Petersen et al., 2021. We use the same codebase and the same hyperparameters. We will clarify this in the camera-ready version of the paper.
>
> We want to emphasize that the main contribution of the paper is not the application of DisCo-DSO to symbolic regression, but rather the investigation of the coupled discrete-continuous optimization approach in generative optimization scenarios where the solution is constructed sequentially. Nevertheless, following the reviewer's suggestion, we have computed results for two recent deep learning approaches to symbolic regression: Kamienny et al., 2022 and Biggio et al., 2021. These are the results that we obtained for a similar number of function evaluations (note that their code does not natively report the number of function evaluations) for Kamienny et al., 2022:
>
> | Model | Mean R test | Std R test |
> | --- | --- | --- |
> | DisCo-DSO | 0.7045 | 0.3007 |
> | Decoupled-RL-BFGS | 0.6400 | 0.3684 |
> | Decoupled-RL-evo | 0.0969 | 0.2223 |
> | Decoupled-RL-anneal | 0.1436 | 0.3015 |
> | Decoupled-GP-BFGS | 0.4953 | 0.4344 |
> | Decoupled-GP-evo | 0.0747 | 0.1763 |
> | Decoupled-GP-anneal | 0.1364 | 0.2608 |
> | Kamienny et al., 2022 | 0.5699 | 0.1065 |
>
> and for Biggio et al., 2021 (note that this system only supports $\leq 3$ dimensions):
>
> | Model | Mean R test | Std R test |
> | --- | --- | --- |
> | DisCo-DSO | 0.6632 | 0.3194 |
> | Decoupled-RL-BFGS | 0.6020 | 0.4169 |
> | Decoupled-RL-evo | 0.0324 | 0.1095 |
> | Decoupled-RL-anneal | 0.1173 | 0.2745 |
> | Decoupled-GP-BFGS | 0.5372 | 0.4386 |
> | Decoupled-GP-evo | 0.0988 | 0.1975 |
> | Decoupled-GP-anneal | 0.1615 | 0.2765 |
> | Biggio et al., 2021 | 0.6858 | 0.1995 |
>
>
> We can see that the coupled approach of DisCo-DSO still outperforms or gives comparable results to these baselines. We will add these results together with a discussion about the differences between DisCo-DSO and these baselines in the camera-ready version of the paper.
>
>
> References:
>
> - John R Koza. Genetic programming as a means for programming computers by natural selection. Statistics and computing, 4:87–112, 1994.\\
> - Brenden K. Petersen et al. Deep symbolic regression: Recovering mathematical expressions from data via risk-seeking policy gradients.  ICLR 2021, Virtual Event, Austria, May 3-7, 2021.\\
> - Pierre-Alexandre Kamienny, St´ephane d’Ascoli, Guillaume Lample, and Franc¸ois Charton. End-to- end symbolic regression with transformers. arXiv preprint arXiv:2204.10532, 2022.\\
> - Luca Biggio, Tommaso Bendinelli, Alexander Neitz, Aurelien Lucchi, and Giambattista Parascan- dolo. Neural symbolic regression that scales. In International Conference on Machine Learning, pp. 936–945. PMLR, 2021.

---

### Official Review · Reviewer_AbyM · 2023-11-03

**Soundness:** 2 fair
**Presentation:** 2 fair
**Contribution:** 2 fair
**Rating:** 3
**Confidence:** 3

**Summary:**

This paper proposes DisCO-DSO, a generative modeling approach to learn a joint distribution over continuous and discrete variables. Prior works follow a decoupled approach, where the discrete and continuous variables are modeled separately, leading to inefficiency in the sampling and optimization procedure. DisCO-DSO uses an autoregressive model and produces two latent variables, one is used to generate the discrete distribution, and the other is used to generate the continuous distribution. This method requires one evaluation step per sample, whereas prior approaches use black-box optimization methods to sample the continuous variable for each discrete token. Experiments are performed on a newly proposed parameterized bitstring task, symbolic regression for equations, and learning decision tree policies for RL. The experiments demonstrate competitive performance with existing methods while improving efficiency.

**Strengths:**

This work proposes a fairly simple approach to model hybrid spaces, which improves the efficiency compared to existing work. The idea of modeling the discrete and continuous distributions using different latent vectors in this context is novel, to the best of my understanding. The parameterized bitstring task is simple yet effective for benchmarking the performance of hybrid space generative models. In my opinion, this work has a moderate impact on a specific sub-area of generative modeling.

The presentation and quality of writing are mostly clear. The paper provides relevant context and then describes the proposed method along with model diagrams to illustrate the difference to prior work clearly.

**Weaknesses:**

The main weaknesses of the paper are poor baselines in the experiments and some organizational changes for clarity. The experiments consider baselines that decouple the discrete and continuous space optimization, but many of the recent works mentioned in the related works are not considered as baselines. Without this comparison, it is difficult to ascertain the empirical performance of DisCO-DSO. Another issue is that while the writing is clear, there are some minor organizational changes that can improve the readability of the paper.  See the questions below for more details.

**Questions:**

**********************Comparison with prior work:********************** The related works section describes prior work in the area with different approaches to the problem of modeling joint discrete-continuous spaces, such as Petersen et al., 2021;  Kamienny et al., 2022; Sahoo et al., 2018 and specifically for symbolic regression such as Biggio et al., 2021; Landajuela et al., 2021. The comparison with Petersen et al., 2021 is especially relevant since DisCO-DSO uses the same risk-seeking policy gradient approach to optimize the reward-based objective. Without comparison with relevant prior work, it is difficult to accurately gauge the significance of the empirical contribution.

****************************************************************Choice of autoregressive model:**************************************************************** DisCO-DSO uses LSTMs for autoregressive sequence generation. It would be interesting to see the effect on performance if the backbone model was changed, possible options include GRUs and Transformers.

********************************************************Organization and structure:******************************************************** The readability of the paper can be improved by using paragraph titles to better organize large bodies of text, particularly Section 2, Section 3.2, Section 4.2 and Section 4.3.

**********************References:**********************

- Brenden K. Petersen, Mikel Landajuela, T. Nathan Mundhenk, Cl´audio Prata Santiago, Sookyung
Kim, and Joanne Taery Kim. Deep symbolic regression: Recovering mathematical expressions
from data via risk-seeking policy gradients. In 9th International Conference on Learning Rep-
resentations, ICLR 2021, Virtual Event, Austria, May 3-7, 2021. [OpenReview.net](http://openreview.net/), 2021a. URL
https://openreview.net/forum?id=m5Qsh0kBQG.
- Pierre-Alexandre Kamienny, St´ephane d’Ascoli, Guillaume Lample, and Franc¸ois Charton. End-to-
end symbolic regression with transformers. arXiv preprint arXiv:2204.10532, 2022.
- Subham Sahoo, Christoph Lampert, and Georg Martius. Learning equations for extrapolation and
control. In International Conference on Machine Learning, pp. 4442–4450. PMLR, 2018. URL
http://proceedings.mlr.press/v80/sahoo18a.html.
- Luca Biggio, Tommaso Bendinelli, Alexander Neitz, Aurelien Lucchi, and Giambattista Parascan-
dolo. Neural symbolic regression that scales. In International Conference on Machine Learning,
pp. 936–945. PMLR, 2021.
- Mikel Landajuela, Brenden K Petersen, Sookyung Kim, Claudio P Santiago, Ruben Glatt, Nathan
Mundhenk, Jacob F Pettit, and Daniel Faissol. Discovering symbolic policies with deep rein-
forcement learning. In Marina Meila and Tong Zhang (eds.), Proceedings of the 38th International
Conference on Machine Learning, volume 139 of Proceedings of Machine Learning Research, pp.
5979–5989. PMLR, 18–24 Jul 2021c. URL https://proceedings.mlr.press/v139/
landajuela21a.html.

---

> ### Author Response · Authors · 2023-11-18
> **Rebuttal by Authors to Reviewer AbyM**
>
> We thank the reviewer for her/his comments. The "summary" and "strengths" sections that the reviewer compiled show that the reviewer has read the paper carefully and understood the main contributions of the paper. We will address the reviewer's comments in the following.
>
> **"In my opinion, this work has a moderate impact on a specific sub-area of generative modeling".**
> We believe that the field of optimization with generative models is a very important area of research with many real-world applications, whose importance is only going to increase in the future. We believe that our method can have a significant impact in this field by providing a rather simple and effective way to optimize generative models in hybrid spaces.
>
> **Comparison with prior work**.
> We want to clarify that the baseline Petersen et al., 2021 is present in each experiment under the alias "Decoupled-RL-BFGS". Our work builds upon the openly available codebase of Petersen et al., 2021, accessible at https://github.com/dso-org/deep-symbolic-optimization, establishing a direct one-to-one correspondence between Petersen et al., 2021, and "Decoupled-RL-BFGS". We regret any lack of clarity in our initial presentation and commit to explicitly addressing this correspondence in the forthcoming camera-ready version.
>
> The reviewer cites baseline studies by Kamienny et al., 2022; Sahoo et al., 2018; Biggio et al., 2021; and Landajuela et al., 2021. Please note that the method in Landajuela et al., 2021 corresponds to Petersen et al., 2021 (but applied to the symbolic policy discovery for RL). Thus, this method is already present in our paper under the alias "Decoupled-RL-BFGS".
>
> We compile results on our validation benchmark for Kamienny et al., 2022 using the codebase provided by the authors. These are the results that we obtained for a similar number of function evaluations (note that their code does not natively report the number of function evaluations, so we had to estimate it):
>
> | Model | Mean R test | Std R test |
> | --- | --- | --- |
> | DisCo-DSO | 0.7045 | 0.3007 |
> | Decoupled-RL-BFGS | 0.6400 | 0.3684 |
> | Decoupled-RL-evo | 0.0969 | 0.2223 |
> | Decoupled-RL-anneal | 0.1436 | 0.3015 |
> | Decoupled-GP-BFGS | 0.4953 | 0.4344 |
> | Decoupled-GP-evo | 0.0747 | 0.1763 |
> | Decoupled-GP-anneal | 0.1364 | 0.2608 |
> | Kamienny et al., 2022 | 0.5699 | 0.1065 |
>
> Following the reviewer's suggestion, we also consider Biggio et al., 2021 as a baseline. However, we were not able to directly apply their method to our validation benchmark, as their system only supports $\leq 3$ dimensions. We then just report the subsets of the validation benchmark that are supported by their system. These are the results that we obtained for a similar number of function evaluations (note that their code does not natively report the number of function evaluations and we had to estimate it):
>
> | Model | Mean R test | Std R test |
> | --- | --- | --- |
> | DisCo-DSO | 0.6632 | 0.3194 |
> | Decoupled-RL-BFGS | 0.6020 | 0.4169 |
> | Decoupled-RL-evo | 0.0324 | 0.1095 |
> | Decoupled-RL-anneal | 0.1173 | 0.2745 |
> | Decoupled-GP-BFGS | 0.5372 | 0.4386 |
> | Decoupled-GP-evo | 0.0988 | 0.1975 |
> | Decoupled-GP-anneal | 0.1615 | 0.2765 |
> | Biggio et al., 2021 | 0.6858 | 0.1995 |
>
>
> We can see that the coupled approach of DisCo-DSO still outperforms or gives comparable results to these baselines (note that the results of Biggio et al., 2021 are only comparable for $\leq 3$ dimensions). We will add these results together with a discussion about the differences between DisCo-DSO and these baselines in the camera-ready version of the paper.
>
> **Choice of autoregressive model**.
> The reviewer raises an interesting point. To address it, we have repeated the experiments in Symbolic Regression (Task 4.2) using different autoregressive models of different sizes. Specifically, we consider a GRU and a LSTM recurrent cell with 16, 32, and 64 hidden units. The results are shown in the following table:
>
> | Model | Mean R test | Std R test |
> | --- | --- | --- |
> | DisCo-DSO-GRU16 | 0.7377 | 0.3161 |
> | DisCo-DSO-GRU32 | 0.7092 | 0.3442 |
> | DisCo-DSO-GRU64 | 0.7236 | 0.3261 |
> | DisCo-DSO-LSTM16 | 0.7385 | 0.3177 |
> | DisCo-DSO-LSTM32 | 0.7241 | 0.3134 |
> | DisCo-DSO-LSTM64 | 0.7302 | 0.3182 |
>
> We observe that the results show little difference between the different models architectures and sizes. This trend is consistent with results in the RL literature, where often the choice size of the policy network has little effect on the performance of the algorithm, as long as the network is large enough to capture the complexity of the problem. We will add this discussion in the camera-ready version of the paper.
>
> **Organization and structure**.
> We will add paragraph titles to the sections mentioned by the reviewer in the camera-ready version of the paper.

---

### Meta-Review · Area_Chair_7L1j · 2023-12-04

**Metareview:**

This paper proposes to add continuous parameter support for a Pointer Network to extend its capabilities beyond pure discrete combinatorial optimization, to more continuous + hybrid search spaces, such as:
* Hybrid, flat search space (Parameterized BitString Task)
* Conditional, tree-based spaces (Symbolic Regression, Decision Tree Policies for RL).

Experiments conducted show that the proposed method outperforms other decoupled methods (i.e. ones which only create the tree, and use a separate solver like L-BFGS to compute continuous constants).

The main issue of the paper currently is one of presentation. There are two main weaknesses that unfortunately, cannot be resolved in time:
* As seen from low novelty comments from reviewers, the writing does not justify why modifying the Pointer Network to accept continuous inputs is substantial. It indeed may be the case that the RL-combinatorics community has overlooked this simple modification which may lead to significant gains over previous work (e.g. Decoupled-RL-BFGS). But it would improve the paper's story much more if their overlooking is explicitly noted.
* The presentations of the experiments are not doing them justice. The results do in-fact show that the proposed DISCO-DSO is performing quite well, but the plots are poorly made (ex: Figure 2 could simply have separate additional plots for when $\alpha=0.5$ rather than dashed lines) that it is hard to tell that the paper has made a significant contribution.

Unfortunately, I believe these are the primary reasons which led to lower reviewer scores, despite the strong inherent experimental results. I strongly recommend polishing the paper's presentation (both conceptually and experimentally) and submit to the next venue.

**Justification For Why Not Higher Score:**

The paper needs to improve its presentation to better highlight its contributions. Currently it needs a significant re-write and is not acceptable given the deadline.

**Justification For Why Not Lower Score:**

N/A

---

### Decision · Program_Chairs · 2024-01-16

Reject